# GCI30: a global dataset of 30-m cropping intensity using multisource remote sensing imagery

Miao Zhang[1], Bingfang Wu[1,2] *, Hongwei Zeng[1,2], Guojin He[1,2], Chong Liu[3] *, Shiqi Tao[4], Qi Zhang[5,6], Mohsen Nabil[1,2,7], Fuyou Tian[1,2], José Bofana[1,2,8], Awetahegn Niguse Beyene[1,2,9], Abdelrazek Elnashar[1,2,10], Nana Yan[1], Zhengdong Wang[1,2], Yiliang Liu[11]

State Key Laboratory of Remote Sensing Science, Aerospace Information Research Institute, Chinese Academy of Sciences, Beijing 100101, PR China
University of Chinese Academy of Sciences, Beijing 100049, PR China
School of Geospatial Engineering and Science, Sun Yat-Sen University, Guangzhou, 510275, PR China
Graduate School of Geography, Clark University, Worcester, MA 01610, USA
Department of Earth and Environment, Boston University, Boston, MA 02215, USA
Frederick S. Pardee Center for the Study of Longer-Range Future, Frederick S. Pardee School of Global Studies, Boston University, Boston, MA 02215, USA
Division of Agriculture Applications, Soils, and Marine (AASMD), National Authority for Remote Sensing & Space Sciences (NARSS), Cairo, New Nozha, Alf Maskan,1564, Egypt
Center for Agricultural and Sustainable Development Research (CIADS), Catholic University of Mozambique-Faculty of Agricultural Sciences, Cuamba 3305, Mozambique
Tigray Agricultural Research Institute, P.O. Box 492, Mekelle 251, Ethiopia
Department of Natural Resources, Faculty of African Postgraduate Studies, Cairo University, Giza 12613, Egypt
National Remote Sensing Center of China, Beijing 100036, PR China

*Correspondence to*: Bingfang Wu (wubf@aircas.ac.cn), Chong Liu (liuc@mail.sysu.edu.cn)

**Abstract.** The global distribution of cropping intensity (CI) is essential to our understanding of agricultural land use management on Earth. Optical remote sensing has revolutionized our ability to map CI over large areas in a repeated and cost-efficient manner. Previous studies have mainly focused on investigating the spatiotemporal patterns of CI ranging from regions to the entire globe with the use of coarse-resolution data, which are inadequate for characterizing farming practices within heterogeneous landscapes. To fill this knowledge gap, in this study, we utilized multiple satellite data to develop a global, spatially continuous CI map dataset at 30-m resolution (GCI30). Accuracy assessments indicated that GCI30 exhibited high agreement with visually interpreted validation samples and *in situ* observations from the PhenoCam network. We carried out both statistical and spatial comparisons of GCI30 with six existing global CI estimates. Based on GCI30, we estimated that the global average annual CI during 2016–2018 was 1.05, which is close to the mean (1.09) and median (1.07) CI values of the existing six global CI estimates, although the spatial resolution and temporal coverage vary significantly among products. A spatial comparison with two satellite based land surface phenology products further suggested that GCI30 was not only capable of capturing the overall pattern of global CI but also provided many spatial details. GCI30 indicated that single cropping was the primary agricultural system on Earth, accounting for 81.57% (12.28 million $km^2$) of the world's cropland extent. Multiple-cropping systems, on the other hand, were commonly observed in South America and Asia. We found large variations across countries and agroecological zones, reflecting the joint control of natural and anthropogenic drivers on regulating cropping practices. As the first global coverage, fine-resolution CI product, GCI30 is expected to fill the data gap for promoting sustainable agriculture by depicting worldwide diversity of agricultural land use intensity. The GCI30 dataset is available on Harvard Dataverse: https://doi.org/10.7910/DVN/86M4PO (Zhang et al., 2020).

**1 Introduction**

The interrelated targets of zero hunger, no poverty, and promoting sustainable agriculture have been collectively recognized as the core sustainable development goals (SDGs) by the United Nations (UN, 2015; Wu et al., 2017; Whitcraft et al., 2019; Hinz et al., 2020). However, 750 million people are currently exposed to severe food insecurity, and the COVID-19 pandemic may have added approximately 100 million people to the total undernourished population in 2020 (FAO et al., 2020). Projections have further demonstrated that from 2010 to 2050, the world's agricultural production must increase by 70–110% to meet the demands caused by increasing populations and changing diets (Tilman et al., 2011). However, intensified agricultural activities have many ripple effects on terrestrial ecosystems, including forest degradation (Morton et al., 2006; Zeng et al., 2018), soil pollution (Lal, 2002; Jankowski et al., 2018), and changes in carbon/water flux seasonality (Gray et al., 2014; Hao et al., 2015), which in turn damage the welfare of human society. To meet the critical human needs for food security and environmental sustainability, it is of major scientific significance to better understand how existing agricultural land resources are utilized, both locally and globally.

Cropping intensity (CI), defined as the number of crop planting and harvesting cycle(s) within a full year (Gray et al., 2014; C. Liu et al., 2020), offers a measure of cropland utilization that has profound implications for closing food production gaps and agricultural intensification (Challinor et al., 2015; Ding et al., 2016; Wu et al., 2018; Waha et al., 2020). CI also plays an essential role in crop modelling that assesses grain yield (Becker and Johnson, 2001), soil quality (Sherrod et al., 2003), and the impacts of climate change (Pielke Sr et al., 2007; Challinor et al., 2015). Given its importance, it is necessary to accurately estimate CI to improve the management of agricultural activities as well as their interactions with other physical components of the Earth system. Before the advent of remote sensing, information about CI could be estimated only based on limited agricultural census data, but these data are often outdated and variable in accuracy (L. Liu et al., 2020). Remote sensing has revolutionized our ability to estimate CI, especially at continental to global scales (L. Liu et al., 2020). The presence of crop growth and senescence phenology constitutes the most characteristic temporal feature of agricultural practices, and numerous attempts have been made to link high temporal frequency vegetation index time series to CI identification. A growing season peak detection-based algorithm was developed and used it to monitor the CI spatiotemporal change in China (Yan et al., 2014; 2019). A similar approach was adopted by Kotsuki and Tanaka (2015) to derive a global crop calendar dataset containing CI metrics. Despite their prominent contributions to cropland intensification assessments, most existing CI products have coarse spatial resolutions, giving rise to the common presence of mixed pixels that can lead to a decreased CI mapping accuracy. To alleviate this issue, in recent years, fine-resolution optical satellite sensors, such as Landsat and Sentinel-2, have been employed to extract CI information. For example, Jain et al. (2013) found that fine-resolution satellite imagery can more accurately depict the CI pattern in smallholder agriculture regions than can coarse-resolution satellite data. Hao et al. (2019) also reported an improved performance of CI identification using harmonized Landsat Sentinel-2 (HLS) data.

It is becoming increasingly clear that a global, fine-resolution CI product is essential for monitoring the ongoing cropland intensification process on Earth. However, to the best of our knowledge, such a dataset has not yet been

created, reflecting the necessity of a generalizable CI mapping framework that is representative of diverse climate zones and cropping systems. To overcome this challenge, we proposed a phenophase-based CI mapping framework in our pilot study (C. Liu et al., 2020) with the use of multiple satellite data and the Google Earth Engine (GEE) platform (Gorelick et al., 2017), offering both methodological and practical bases for operationalizing a global fine-resolution CI product. Taking advantage of the proposed framework, the primary goal of this research is to advance and develop a global, spatially continuous CI map at a 30-m resolution (GCI30). To achieve this goal, we regenerated the global cropland extent layer and modified the CI estimation algorithm on flooded rice paddies by considering the flooding/transplanting signals. We integrated the full archive of Landsat, Sentinel-2 and MODIS data from 2016 to 2018 for constructing seamless spectral time series in order to capture the full cropping cycles, which is the key for CI identification by segmenting growing and non-growing periods. The performance of GCI30 was examined with *in situ* data as well as with six existing global products containing CI metrics. With a much finer spatial resolution and global coverage, GCI30 is expected to contribute to our fundamental understanding of the dynamics of the Earth's terrestrial surface as well as the human role in land modification through agricultural activities.

## 2 Materials and Methods

### 2.1 GCI30 input data

#### 2.1.1 Cropland extent

The cropland definition adopted in this study is based on the concept presented by the Joint Experiment of Crop Assessment and Monitoring (JECAM) network, which uses a shared definition of the cropland that matches the Food and Agriculture Organization's (FAO) Land Cover Meta Language. The annual cropland (including area affected by crop failure) is defined as a piece of arable land that is sowed or planted at least once within a 12-month period (Waldner et al., 2016). One exception to the above-mentioned cropland definition is the fields used for sugarcane and cassava cultivation, which are included in the cropland class although they have a longer vegetation cycle and are not annually planted and harvested. We integrated ten existing land cover maps or cropland datasets to delimit the global cropland extent while masking out irrelevant non-cropland pixels for the period of 2016–2018 (Figure 1). Readers can refer to Table S1 for detailed information on these land cover/cropland layer products as well as their classes used in the integration. Although variations of classification systems among different products exist, a subset classes of those land cover/cropland layer products were selected to best fit into the cropland definition. Spatially, FROM-GLC was selected for Europe, Africa, New Zealand, the majority of Asia, and part of Latin America. GFSAD30 was selected for tropical Asian islands, including Indonesia, Malaysia and the Philippines. In addition to these two global-coverage cropland extent products, several national or regional datasets, including ChinaCover, CDL, AAFC ACI, NLCD, MapBiomass, CLUM, SERVIR and INTA, were used because they have been extensively validated by local experts and hence exhibited high accuracies of cropland mapping. Their spatial extents cover China, the contiguous U.S., Canada, Alaska, Brazil, Australia, the Lower Mekong River basin (Myanmar, Thailand, Lao, Cambodia and Vietnam), and part of Argentina, respectively.

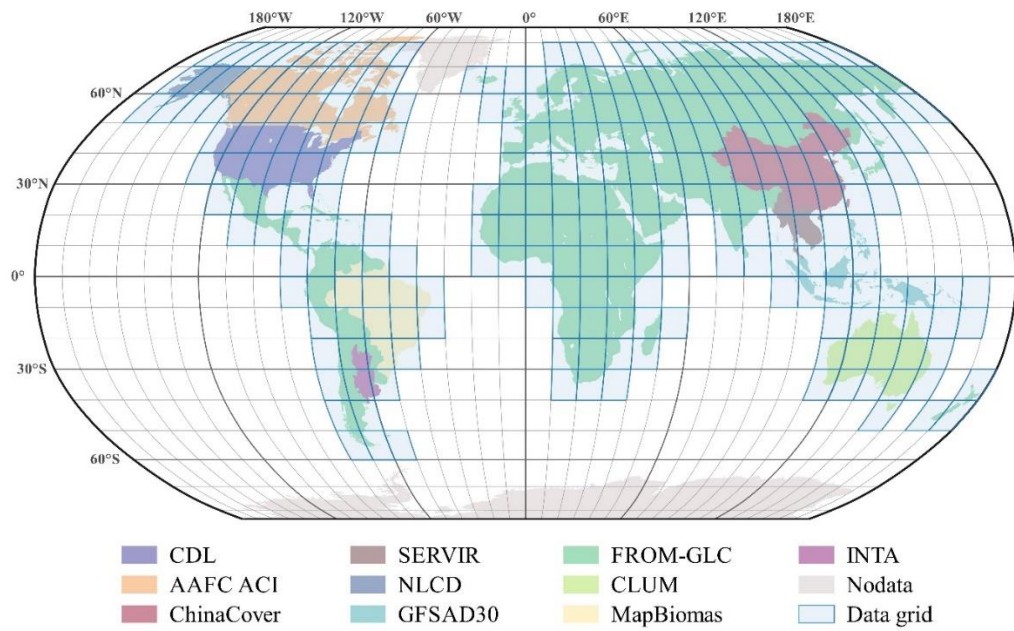

**Figure 1: Spatial distribution of the land cover/cropland layer products used for the global 30-m cropland extent generation.**

### 2.1.2 Satellite images and vegetation indices

All available images of top-of-atmosphere (TOA) reflectance from Landsat‑7 ETM+, Landsat‑8 OLI and Sentinel-2 MSI during 2016–2018 were used for global CI mapping via the GEE platform. Invalid observations, including clouds, cloud shadows, snow and saturated values, were identified and masked by the function of the mask (Fmask) algorithm (Zhu and Woodcock, 2012; Qiu et al., 2019). To overcome the multi-sensor mismatch issue, we adopted an inter-calibration approach, which converted Sentinel-2 MSI and Landsat-8 OLI TOA reflectance data to the Landsat-7 ETM+ standard (Chastain et al., 2019). Then the calibrated images were used to composite the 16-day TOA reflectance time series (labelled as origin).

Based on the harmonized TOA reflectance composite, the following vegetation indices were calculated:

$$NDVI = \frac{\rho_{NIR} - \rho_{RED}}{\rho_{NIR} + \rho_{RED}}$$

$$EVI = 2.5 \times \frac{\rho_{NIR} - \rho_{RED}}{\rho_{NIR} + 6 \times \rho_{RED} - 7.5 \times \rho_{BLUE} + 1}$$

$$LSWI = \frac{\rho_{NIR} - \rho_{SWIR}}{\rho_{NIR} + \rho_{SWIR}}$$

where $\rho_{BLUE}$, $\rho_{RED}$, $\rho_{NIR}$, and $\rho_{SWIR}$ are the TOA reflectance values of the blue, red, near-infrared, and shortwave-infrared bands, respectively. We also used the MOD13Q1 NDVI/EVI and MOD09A1-derived LSWI products (Collection 6) to fill data gaps caused by the vacancy of Landsat/Sentinel-2 observations that were masked out by the

Fmask algorithm. In particular, the coarse MODIS datasets were resized to 30-m using the bicubic interpolation
method. Then an empirical linear function was built for each pixel to bridge the data records of MODIS and
Landsat/Sentinel-2, and missing data gaps were filled with the resampled, transformed MODIS data (labelled as
MODIS modelled). If there is no valid data from either Landsat/Sentinel-2 or MODIS, temporally adjacent (within
48-day) cloud free LANDSAT/Sentinel-2 observations were used to determine the filling value (labelled as
interpolated). After gap-filling, a weighted Whittaker smoother  was further adopted to smooth the gap filled time
series data. We assigned different weights (1, 0.5, 0.2) to Landsat/Sentinel-2 original observations, MODIS modelled
values and interpolation values, respectively. Finally, a dataset of smoothed, seamless image time series of vegetation
indices was generated at a spatial resolution of 30-m with a temporal interval of 16-day.
**2.2 Reference samples**
The validation of the global CI product requires carefully constructed reference samples (Kontgis et al., 2015; Li et
al., 2014; C. Liu et al., 2020). In this study, we constructed two independent reference datasets (termed RDsat and
RDsite hereafter) (Figure S1) to evaluate the GCI30 performance because currently no other product is capable of
offering comprehensive CI mapping assessment at the global scale. The first dataset, RDsat, was generated based on
a visual interpretation of satellite time series via the Geo-Wiki platform (http://www.geo-wiki.org/). Based on the
global segmentation of agroecological zones (termed AEZs here after) (Gommes et al., 2016; 2017) (Table S2), we
applied a stratified sampling approach to ensure that RDsat was geographically representative across the globe. We
divided all 65 AEZs into four categories based on their cropland proportions: VL (cropland proportion < 4%), L (4%
≤ cropland proportion < 15%), M (15% ≤ cropland proportion < 40%) and H (cropland proportion ≥ 40%). For each
category, 1000 plots (each equivalent to a 30-m Landsat pixel size) were randomly collected only within the cropland
extent, and their phenological cycles during the period of 2016–2018 were visually counted. We have seven remote
sensing experts (listed as co-authors) checked all collected plots, and only well-interpreted plots with high-level
confidence were kept, leading to 3744 sample records. The second dataset, RDsite, was derived from PhenoCam
dataset (Richardson et al., 2018a; 2018b; Seyednasrollah et al., 2019), which has been widely used as a robust *in situ*
reference for remotely sensed phenology metric validation. Globally, there are 115 PhenoCam sites on cropland, and
a total of 40 sites were collected after removing those with data records of less than one year (Table S3). For each
selected site, we used the green chromatic coordinate (GCC) index (Richardson et al., 2018b) and *in situ* phenology
camera image time series for cropping cycle number identification. It should be noted that not all the selected
PhenoCam sites precisely covered a period matching 2016-2018. For instance, the site with ID of 'usof6' in Table S3
provided measurements from May 2018 to September 2020, which was out of the study period used for our GCI30
product. However, to make full use of these measurements, we implemented our approach of CI identification by
aligning the study period with the period containing the measurements at each of these sites. Thus, some CI
identification covered longer periods than the three-year length (2016-2018).

## 2.3 Global cropping intensity mapping method

### 2.3.1 CI mapping on non-flooded croplands

We applied the framework designed by C. Liu et al. (2020) for mapping global CI at the 30-m spatial resolution, using the phenophase-based approach for the non-flooded cropland and improving the algorithm to consider flooded rice paddy (section 2.3.2). Targeting non-flooded cropland pixels, the methodology can be divided into two main steps, including i) the identification of subperiod for a complete phenological phase of cropland within each cropland pixel, and ii) the derivation of cropping cycles for CI mapping.

With the smoothed, seamless NDVI time series, we identified a complete phenological cycle of crop via detecting the transitioning points that characterize the phenophase. Within the entire study period, 2016-2018, the transitioning points were located as the 50% of the NDVI amplitude (i.e., different between the minimum and maximum values of observations) and labelled either greening-up points or greening-down points (Bolton et al., 2020). A point at the position where the slope of the time-series function is positive is a greening-up point, while a point at the negatively changing time series is a greening-down point. A pair of points of these two types can separate NDVI time series into one staggered segment consisting of a growing sub-period and a non-growing sub-period.

Based on the segmented time series, we determined the potential number of complete phenophase cycles ($N_{pc}$) by taking the minimum value of the total numbers of transitioning points between the two types (greening-up and greening-down, labelled $N_{up}$ and , $N_{down}$, respectively) within the study period, formulated as:

$$N_{pc} = \min\{N_{up}, N_{down}\}$$

False cycles may exist due to the outliers of NDVI observations, with falsely detected cycles characterized by unrealistically short sub-periods pertaining to crop phenology (Yan et al., 2019). In other words, it is practically impossible for crop to be grown and harvested within a rather short time. In this case, we set a lower limit of the growing-harvesting cycle length to 48 days for removing the false cycles, noted as $N_{fc}$. By adjusting this falsely detected cycles and considering the three-year study period (2016-2018), we calculated the annual cropping intensity (CI) for a non-flooded cropland pixel as:

$$\text{CI} = \frac{N_{pc} - N_{fc}}{3}$$

It should be noted that using the binary phenophase profile itself is not effective enough for identifying the continuous cropping type, which is defined as cropping systems having short growing period (CI > 3 for this study) or exhibiting a lower degree of seasonality (e.g., tea plantation). Therefore, for each cropland pixel, we calculated the coefficient of variation (the ratio of the standard deviation to the mean, termed CV hereafter) of the NDVI time series and adopted a threshold method to determine whether the pixel belonged to the continuous cropping type (low CV value). Specifically, within each AEZ, half of the RDsat samples labelled continuous cropping (if they existed) were adopted

to obtain the CV threshold. This method generated an independent continuous cropping type layer, which was
integrated with the initially derived CI result.
**2.3.2 CI mapping on flooded rice paddies**
Flooded rice paddy, which accounts for more than 12% of the global cropland area and feed approximately half of the
population (FAOSTAT, 2010; Ding et al., 2020), bear special mention in our study because it supports the only staple
grains that need to be transplanted (Dong and Xiao, 2016), resulting in a relatively short non-growing period that may
be mistakenly missed when using the abovementioned approach. This issue becomes more prominent in areas with
high cloud cover (e.g., Monsoon Asia). Therefore, we modified our approach in flooded rice paddy areas by
considering the influence of the "flooding/transplanting signal" on the created phenophase profile (Figure 2). Similar
to the approach for non-flooded croplands, the NDVI time series trajectory was used to generate an initial phenophase
profile (Figure 2a). Then, within each identified growing season, the flooding/transplanting signals were detected and
recognized based on the following criteria: LSWI > EVI or LSWI > NDVI (Figure 2b), indicating that the water signal
dominates the pixel spectral performance (Xiao et al., 2005; Dong et al., 2015; 2016). We regard the
"flooding/transplanting" period as a non-growing phenophase. Thus, the initial phenophase profile can be divided into
two segments accordingly (Figure 2c), reflecting the reality of double-season rice planting cycles. Finally, the CI
information was determined by enumerating the transition points between different cropping cycles. Due to data
limitations, this specific CI identification approach was applied only in southern China (AEZs C33, C37, C40, C41,
and C42) and the Lower Mekong River basin, where the paddy rice type was included in the land cover type scheme
(derived from ChinaCover and SERVIR, respectively).

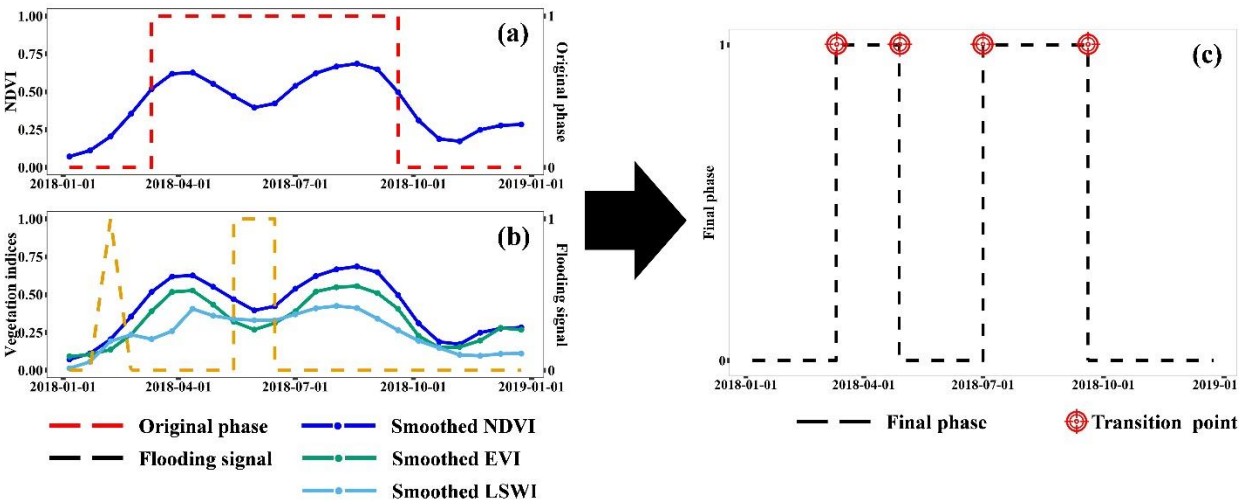

**Figure 2: Illustration of the specific CI identification for a flooded rice paddy pixel.**
**2.4 Accuracy assessment**
Based on the RDsat and RDsite datasets, the accuracy assessment of GCI30 was conducted in two ways. In the first
validation method, we directly evaluated the difference between the reference and estimated results. Here, the total

1     number of cropping cycles (termed TNCC hereafter) rather than the actual CI value was used to avoid decimals. Four

2     complementary indicators, systematic error (*SE*), mean absolute error (*MAE*), root mean square error (*RMSE*) and

3     coefficient of determination ($R^2$), were calculated as follows:

$$SE = \frac{1}{N} \sum_{i=1}^{N} (\hat{f}_i - f_i)$$

$$MAE = \frac{1}{N} \sum_{i=1}^{N} |\hat{f}_i - f_i|$$

$$RMSE = \sqrt{\frac{1}{N} \sum_{i=1}^{N} (\hat{f}_i - f_i)^2}$$

$$R^2 = 1 - \frac{\sum_{i=1}^{N} (\hat{f}_i - f_i)^2}{\sum_{i=1}^{N} (\hat{f}_i - \bar{f})^2}$$

where $\hat{f}_i$ and $f_i$ are the estimated and reference cropping cycle number(s) for sample pixel $i$, respectively. $N$ represents
the number of samples, and $\bar{f}$ is the mean cropping cycle number value of all samples. In addition to directly
quantifying the mapping errors, we further reclassified the GCI30 result into four categories: single cropping ($0 < $ CI
$\leq 1$), double cropping ($1 < $ CI $\leq 2$), triple cropping ($2 < $ CI $\leq 3$) and continuous cropping. We obtained the confusion
matrix and calculated quantitative metrics, including overall accuracy (OA), kappa coefficient, producer accuracy (PA)
and user accuracy (UA). Due to the limited sample sizes of RDsite, the classification-based accuracy assessment was
conducted only for RDsat.
A systematic uncertainty analysis was further applied by interpolating the estimation biases from RDsat samples to a
spatial distribution map (H. Liu et al., 2020). First, the linear normalization was conducted to transform the estimation
bias range of RDsat samples to $[0, 1]$. Then the uncertainty map of GCI30 was created based on the Kriging
interpolation method (Oliver and Webster, 1990). We used the ArcMap software of version 10.1 to implement the
Kriging interpolation in this study, with the spatial search radius parameter set as 12 nearest sample units. As a result,
the values of generated uncertainty map ranges from 0 to 1. A smaller value indicates higher estimation reliability,
while a higher value suggests a greater level of overestimation or underestimation on cropping cycle(s).
**2.5 Comparison with other global products**
Comparison of GCI30 with other global products or studies was conducted statistically and spatially (if available) at
multiple levels. At the global level, we compared and evaluated the statistical differences between GCI30, and six
existing statistical-based or satellite-based products (Table 1), including NASA's Vegetation Index and Phenology
V004 (VIP4) dataset (Didan and Barreto, 2016), MODIS Land Cover Dynamics (MCD12Q2) Version 6 (Gray et al.,

2019), SAtellite-derived CRop calendar for Agricultural simulations (SACRA) (Kotsuki and Tanaka, 2015), harvest

frequency by Ray and Foley (2013) (R&F), actual cropping intensity (ACI) (Wu et al., 2018) and cropland use

intensity (CUI) (Siebert et al., 2010). Among these products, four of them (MCD12Q2, VIP4, SACRA and R&F) were

further employed for national level comparison. It should be noted that comparisons were only conducted for countries

where both GCI30 and reference products are available. Finally, MCD12Q2 and VIP4 were used for pixel by pixel

comparison against GCI30 due to their relatively fine spatial resolutions. To minimize uncertainty caused by temporal

disagreement, we selected only the 2014 VIP4 and the 2016–2018 averaged MCD12Q2 data, within which the

"Number of Seasons" layer of VIP4 and the "NumCycles" layer of MCD12Q2 were extracted for intercomparison.

We upscaled GCI30 to 0.05° and 500 m using the majority algorithm to match the spatial resolution of VIP4 and

MCD12Q2, respectively. The same reclassification procedure described in Section 2.4 transformed the actual CI value

of GCI30 and MCD12Q2 to match the VIP4 dataset's integer value range (0, 1, 2, 3), except for the continuous

cropping and non-cropland pixels that were excluded from our comparison. We generated the difference maps among

GCI30, VIP4 and MCD12Q2 to understand the overall overestimation or underestimation of our mapped CI results

across continents.

**Table 1: Existing cropping intensity products or studies used for inter-comparison.**

| Existing products and studies | algorithm | Input data | Temporal range used | Spatial resolution | Way of comparison |
|---|---|---|---|---|---|
| MCD12Q2 | Phenometrics-based method | Time series MODIS EVI2 | 2016 to 2018 | 500-m | (1) global level; (2) national level; (3) pixel level |
| VIP4 | Half-maximum VI approach | Time series NDVI and EVI2 derived from both AVHRR and MODIS | 2014 | 0.05° | (1) global level; (2) national level; (3) pixel level |
| SACRA | Peak-counting considering crop calendar | Time series of SPOT-VEGETATION NDVI | 2004-2006 | 5′ | (1) global level; (2) national level |
| R&F | The ratio of harvested cropland to total cropland | FAO statistics | 2000-2011 | National level | (1) global level; (2) national level |
| ACI | Peak-counting | Time series GIMMS NDVI3g | 2009-2011 | 8-km | (1) global level |
| CUI | The ratio of harvested crop areas to total cropland area by excluding fallow land | MIRCA crop areas | 2000 | 5′ | (1) global level |

**3 Results and Discussion**
**3.1 Reliability of GCI30**
We examined GCI30 performance by generating a scatter plot of estimated and reference TNCC derived from RDsat
and RDsite, respectively (Figure 3). In general, GCI30 could provide reliable estimation results across different agro-
environmental and management conditions, with relatively small *MAE* and *RMSE* values (equal to or less than 0.4 and
0.92, respectively) using the two reference datasets. Referring to $R^2$, GCI30 captured over 91% of the variation in
RDsite-derived TNCC and over 56% of the variation in RDsat-derived TNCC. The discrepancies between RDsat-
derived metrics and RDsite-derived metrics were mainly attributed to the differences in sample size and crop planting
diversity. Specifically, the network of PhenoCam spots is spatially sparse, and most cropland sites are distributed in
the United States featuring single cropping systems (Figure S1). There were slight systematic underestimations
(negative *SE*) for GCI30, with the larger bias level occurring for RDsite. This result is consistent with C. Liu et al.
(2020), indicating an overall conservative CI estimation by GCI30. In addition, we found that larger estimation errors
were commonly observed in samples with more cropping cycles. This tendency was not surprising because of the
accumulated errors from every aspect of information extracted from remote sensing (Defourny et al., 2019), including
data acquisition, time series modelling of vegetation indices, and phenological cycle identification. Therefore, we may
expect that GCI30 faces larger challenges in terms of analysing multiple cropping systems.

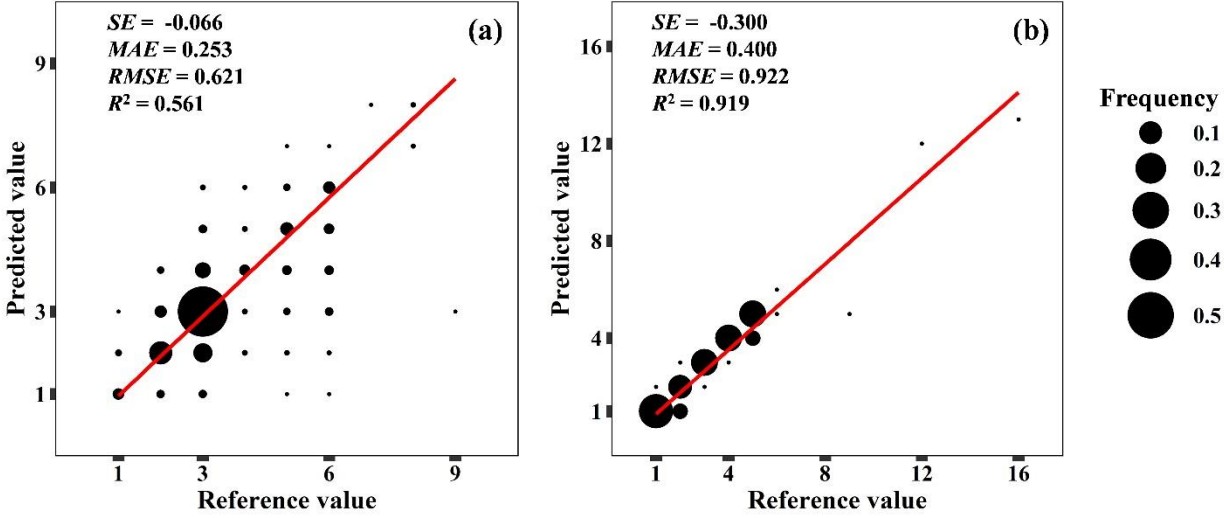

**Figure 3: GCI30 accuracy assessment based on RDsat (a) and RDsite (b). The red line represents the linear fitting line with**
**the intercept forced to 0. The frequency of a specific reference-prediction value pair is proportional to its point size. Samples**
**identified as continuous cropping types were excluded.**
Figure 4 further displays the spatial pattern of RDsat-based TNCC estimation bias across the globe. From 2016 to
2018, 79.8% of the points exhibited unbiased predictions. Among the pixels with disagreement (i.e., non-zero bias),
the majority were associated with one or two cropping cycle difference(s). Overall, there were more underestimation
points (12.2%) than overestimation points (8.0%). Spatially, negative biases were mainly distributed in high CI regions,
including the Pampas (AEZ C26), Central Eastern Brazil (AEZ C23), the Gulf of Guinea (AEZ C03), East African

Highlands (AEZ C02), South of Himalaya (AEZ C44), and Huanghuaihai Plain (AEZ C34), altogether forming a northward "underestimation belt" along the longitudinal gradient. The negative errors could possibly be due to the complexity of some special cropping systems that cannot be fully accounted by our CI mapping method. For example, inter/mixed cropping may lead to shallow troughs in NDVI time series, which makes the 50% NDVI amplitude rule less reliable (C. Liu et al., 2020). Given the conservative CI estimation algorithm, it was somewhat unexpected to observe overestimation errors primarily concentrated in Western Europe (AEZ C60) and Ukraine to the Ural Mountains (AEZ C55), where the single cropping practice dominates due to limited hydrothermal conditions (Wu et al., 2018). These positive biases could be attributed to the fallow strategy adopted in some Europe countries (Estel et al., 2016). During a fallow cycle, there may exist weeds which are falsely identified as one solid cropping cycle. In summary, the global bias distribution highlights the complex suite of biotic and abiotic processes that can obscure the effectiveness of the phenophase-based CI mapping framework.

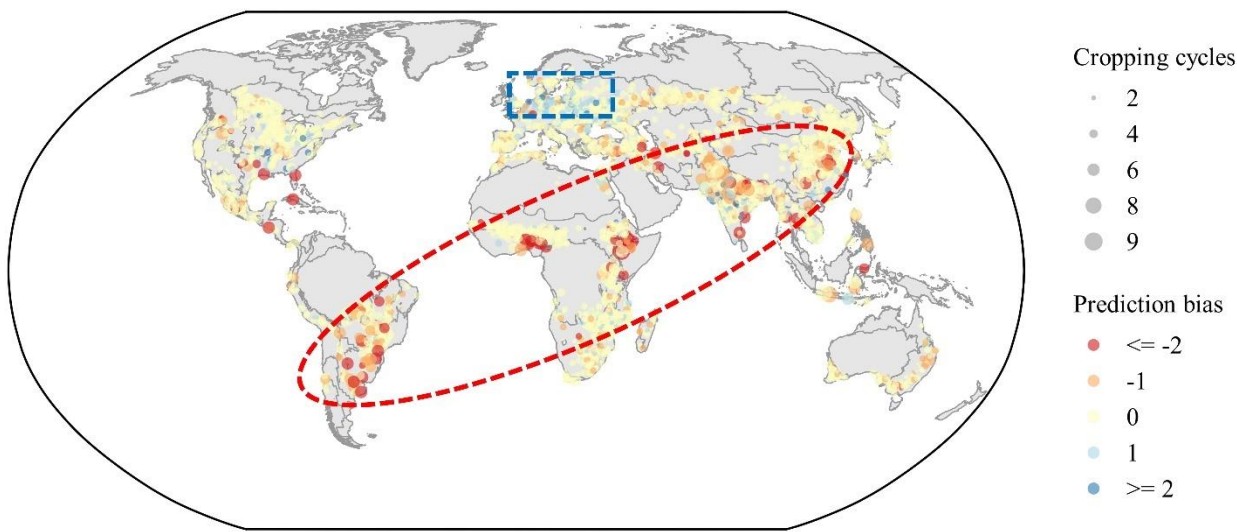

**Figure 4: Spatial distribution of RDsat-based TNCC bias. The actual TNCC is proportional to its point size, and the prediction biases are identified by different point colours. Sample points identified as the continuous cropping type were excluded. The blue rectangle indicates overestimations, and the red ellipses region represents the northward "underestimation belt" along the longitudinal gradient.**

Following the reclassification procedure illustrated in Section 2.4, we derived the corresponding confusion matrix of GCI30 using RDsat samples, with the quantitative accuracy metrics shown in Table 2. We found that GCI30 had reasonable classification performances, with OA and kappa coefficients greater than 92% and 0.72, respectively. Regarding the classes of CI, single-cropping systems were associated with more robust classification results than were multiple-cropping systems. Comparatively, the single-cropping class was more subject to commission errors than omission errors (PA > UA), while the opposite tendency (PA < UA) was observed for the double- and triple-cropping classes. Continuous cropping is a fundamentally different agricultural land use management type from others. Here, we found that the continuous cropping class of GCI30 had a higher PA (93.1%) than UA (77.0%). A possible explanation for this result is likely attributed to the threshold-based method for continuous cropping identification. Some noncontinuous cropping systems may also exhibit low CV values, leading to a relatively high commission error

level. Notably, although a stratified sampling strategy was conducted for creating RDsat, its sample size was still unbalanced among the different CI classes. The single-cropping class alone occupied 88.9% of the total number of samples. Therefore, future efforts of GCI30 validation need to emphasize the inclusion of more samples with multiple-cropping systems.

**Table 2: Confusion matrices of GCI30.**

| | CI class type | Single | Double | Triple | Continuous | UA (%) |
|---|---|---|---|---|---|---|
| | Single | 3059 | 186 | | | 94.3 |
| | Double | 52 | 346 | 2 | 3 | 85.9 |
| **Reference** | Triple | 1 | | 6 | 2 | 66.7 |
| | Continuous | 9 | 9 | 2 | 67 | 77.0 |
| | PA (%) | 98.0 | 64.0 | 60.0 | 93.1 | |
| | OA (%) | 92.9 | | | | |
| | Kappa | 0.728 | | | | |

A spatial distribution map of the GCI30 uncertainty was generated based on the RDsat. As shown in Figure 5, most cropland areas display blue tones, indicating their low uncertainties and high estimation reliabilities of the GCI30. Spatially, relatively lower uncertainty levels were observed in Canada, North Plains in the United States, Russia, northern China, and southern Africa, where single cropping dominates. Meanwhile, there are still some areas of cropland with higher uncertainty as illustrated in orange and red tunes, mostly distributed at Pampas, Southern Nigeria, Central and Northern India.

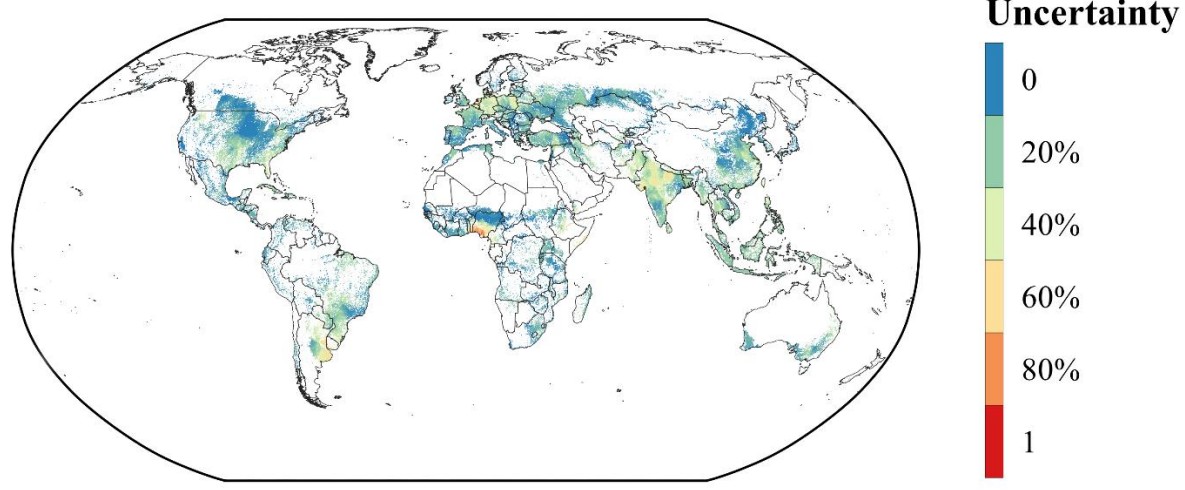

**Figure 5: Global uncertainty map of GCI30 during 2016-2018, where regions in red represent higher uncertainty and those in blue represent lower uncertainty.**

**3.2 Spatial pattern of GCI30**

GCI30 provides the first spatially continuous map of global CI at a 30-m resolution (Figure 6). Based on this map, a heterogeneous pattern in CI compositions across continents was found, which are subject to varying anthropogenic

and climate conditions. Overall, as expected, single cropping was the primary agricultural system on Earth, accounting for 81.57% (12.28 million km$^2$) of the world's cropland extent. Double cropping, on the other hand, was typically implemented in Asia, South America and the Nile River basin of Africa, together occupying 17.42% (2.62 million km$^2$) of global croplands. Comparatively, the proportions of triple and continuous cropping were quite small, with their distributions mainly limited to Southeast Asia. According to the area statistics at five-degree intervals, we found that the area of single cropping reached 54% or higher in all latitude and longitude zones. The double cropping distribution along latitude peaked in intervals ranging from 20°N to 40° N, which encompassed China and India, the two most populous countries in the world. Along longitude, double cropping was mainly concentrated in three zones: 55°W to 60°W, 75°E to 90°E and 100°E to 125°E. These regions are commonly characterized by warm and humid climates, except for the Nile River basin, in which irrigation has been commonly used to support intensive farming practices (Zohaib and Choi, 2020). Over 75% of triple and continuous cropping areas are located within tropical zones (5°S to 5°N). The tropical rainforest climate of these regions ensures sufficient water and heat supplies for crop growth throughout the year (Köppen et al., 2011).

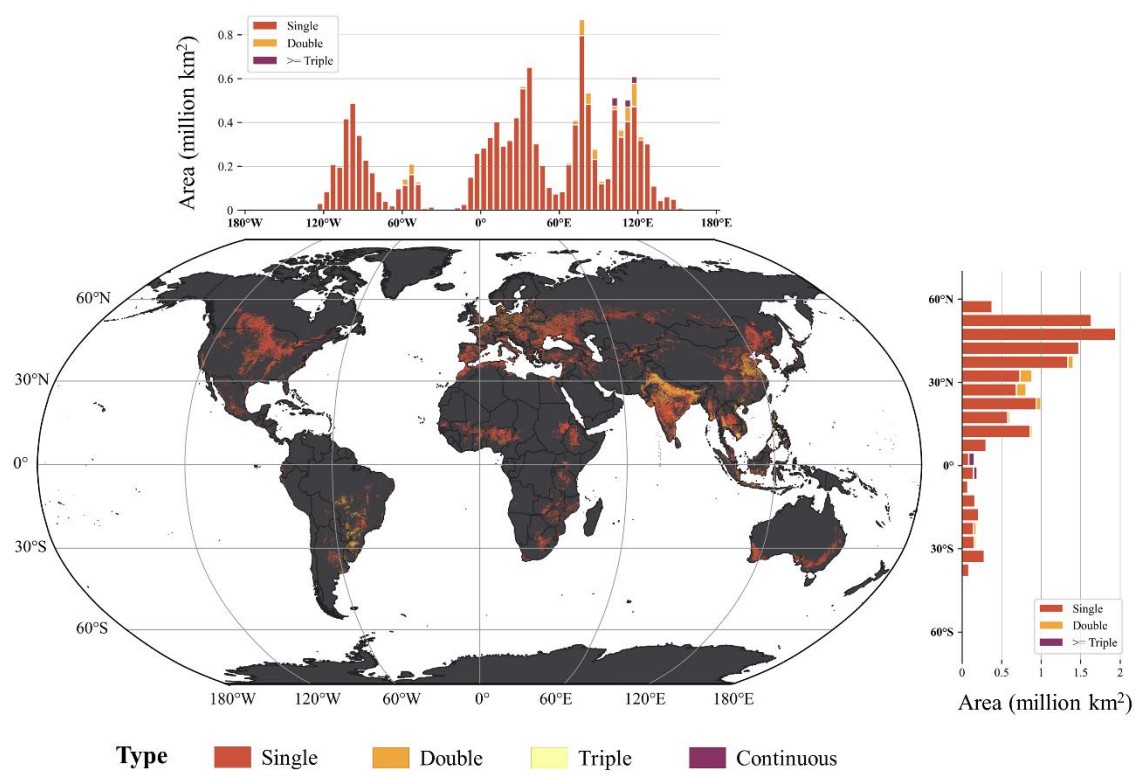

**Figure 6: Geographical distribution of global CI types during 2016 to 2018 identified by GCI30. The area statistics along latitude and longitude are derived with an interval of five degrees. The area unit is million km$^2$.**

Figure 7 displays the GCI30-based TNCC statistics at the continent level. We combined Australia and Oceania (New Zealand, Melanesia, Micronesia and Polynesia) due to the rarity of cropland on these two continents. Globally, South America exhibited the most intensified cropping level, followed by Asia and Europe. Specifically, the average TNCC

values were 3.67, 3.38 and 3.07 for South America, Asia and Europe, respectively. South America and Asia also possessed the largest standard deviations of TNCC, indicating the inherent diversity of agricultural activities within these two continents, as weather conditions directly affect cropping practices (Iizumi and Ramankutty, 2015). For example, in Asia, triple and continuous cropping systems were distributed in Southeast Asia, including Indonesia, Malaysia, southern Thailand and the Mekong River Delta in Vietnam. Double cropping was concentrated in the North China Plain, Ganga River basin and southern China, while the rest of Asia was dominated by a single cropping pattern, covering Central Asia, Northeast Asia, and southern India. Following these continents, moderate TNCC was found in North America (2.93±0.54) and Africa (2.78±0.71). Among all continents, the lowest TNCC occurred in Australia and Oceania (2.31±0.77), where arid and semiarid climate types are dominant (Köppen et al., 2011; Beck et al., 2018).

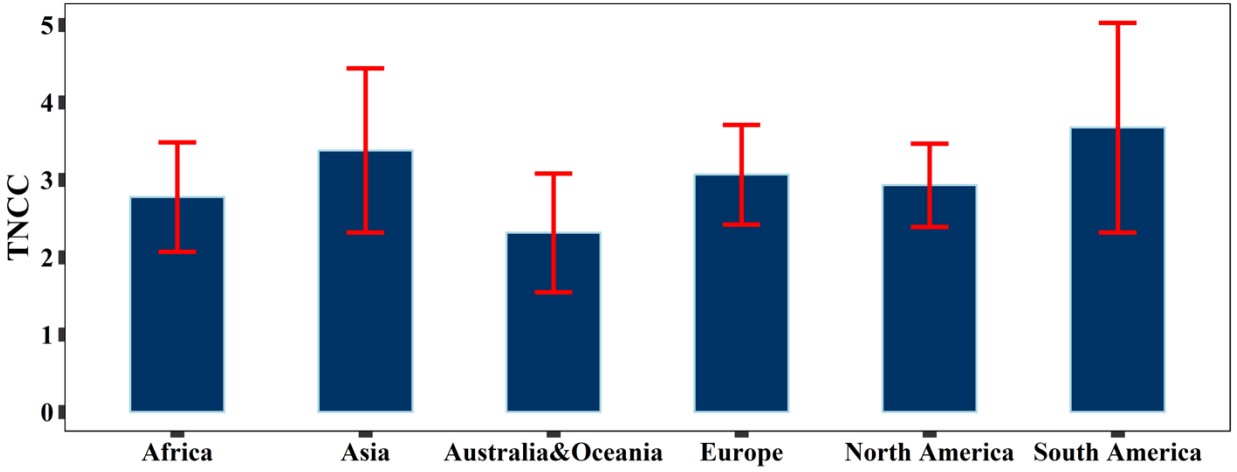

**Figure 7: Statistics of GCI30-based TNCC during 2016 to 2018 at the continent level. The red line indicates the standard deviation.**

At the global scale, the average CI pattern was heterogeneous across countries and AEZs (Figure 8, left panel). Countries with the highest average CI levels were commonly detected in Asia (Bangladesh, Vietnam, Philippines, Sri Lanka) and Latin America (Guyana, Paraguay, Suriname, Haiti, and Dominica). Together with Egypt, these top 10 countries exhibited TNCC values greater than 4.1 during 2016–2018. In contrast, low to moderate CI levels were typically found in high-latitude countries, such as Canada, Russia, and Mongolia. In addition to the latitude gradient, we found that the diversity of cropland management played a critical role in shaping the CI pattern. For example, some high-latitude European countries (Germany, Poland, Belarus, etc.) showed relatively high CI levels due primarily to their advanced cropland management practices (Guo, 2021). Rainfed agricultural practices lead to fewer cropping cycles in the Middle East and North African countries, except for Egypt, where most croplands are irrigated (Wu et al., 2018). Taking climate conditions into account, the heterogeneity of global CI becomes even more prominent among different AEZs. Arid regions, which cover vast areas in Africa, Australia, and Central Asia, are associated with fewer cropping cycles due to a lack of water for irrigation (Chiew et al., 2011; Guo et al., 2018) and less developed agricultural infrastructures (Mason-D'croz et al., 2019). In contrast, intensive farming is widely distributed in humid and low-latitude areas such as South China and the Mekong Delta. Among all 65 AEZs, the

Huanghuaihai Plain in China had the highest CI, followed by the Amazon rainforest region of South America and
Taiwan Province. The lowest CI occurred in the Australian Desert, with an average TNCC less than 2.
Overall, countries and AEZs with intensive farming are more subject to internal variability, as reflected by higher
standard deviations (Figure 8, right panel). Globally, there are 14 countries and 7 AEZs exhibiting standard deviations
greater than 1.2, and most of them are located in South America and Asia. Regions with low CI averages but high CI
standard deviations were observed only on the western coast of South America and Queensland to Victoria in Australia,
where partial irrigation in the former (Xie et al., 2019) and unstable rainfall in the latter resulted in diversified cropping
intensities among years (King et al., 2020). The high standard deviations in Australia and Oceania mainly resulted
from the high within-country/zonal heterogeneity, which may encompass aspects including the exceptionally variable
climate with the prevalence of floods and droughts (King et al., 2020) and the annual shifting of crop types, as well as
cultivated and fallow lands (Song et al., 2017). In addition to these regular drivers, the political situation may cause
CI spatiotemporal diversity. Notably, for instance, we found an unusually high standard deviation in Afghanistan,
which was caused by both crop failure during the emergence to early development stages due to adverse weather
conditions (Rousta et al., 2020) and abandoned cropland resulting from armed conflicts and refugee migrations (Iqbal
et al., 2018; Galdo et al., 2020).

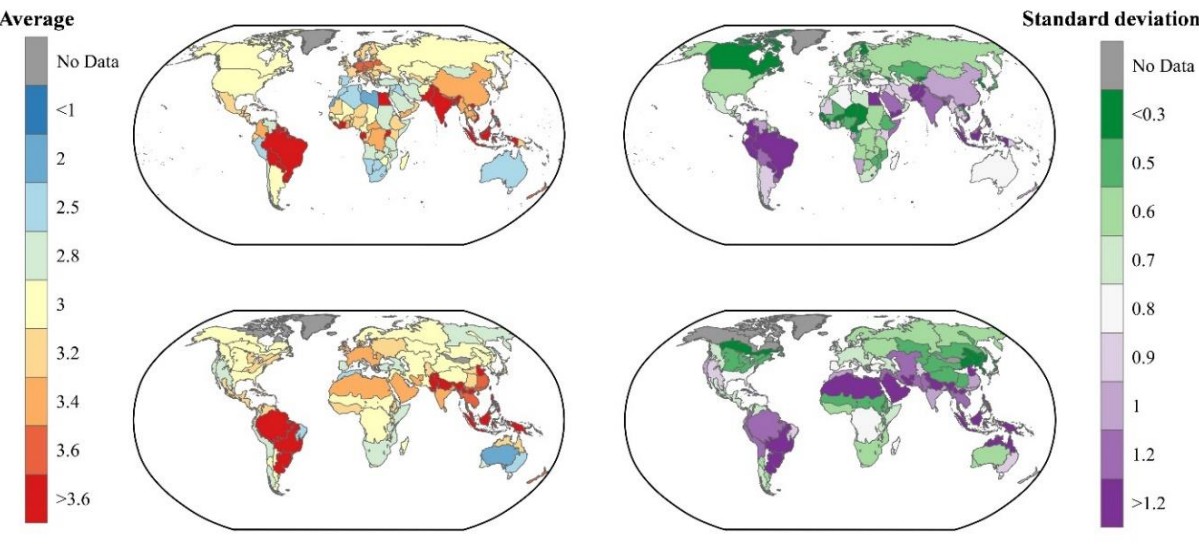

**Figure 8: Average and standard deviation (std) of TNCC during 2016 to 2018 at the national and AEZ levels.**
**3.3 Cross comparison with other studies**
Due to the differences in methods, input data, spatial resolution of the existing products containing CI metrics as listed
in table 2 and GCI30, the statistical average CI at global scales varied among the products. Based on GCI30, the global
average CI during 2016–2018 was 1.05 (the continuous cropping pixel excluded). Statistically, our CI is in a
remarkably high agreement level with estimates based on the existing six estimates (mean CI: 1.09, median CI: 1.07),

despite their significantly varying spatial resolution and temporal coverage. The minimum CI among the seven studies was estimated to be 0.84 per year based on the total cropland extent and the total harvested crop area reported by the agricultural statistics database of the United Nations Food and Agriculture Organization (FAO) FAOSTAT (http://www.fao.org/faostat/en/) (Siebert et al., 2010). CI estimated by GCI30 is slightly higher than that derived from the FAO statistical database. The CI of other existing products listed in table 2 ranges from 1.05, as estimated from "NumCycles" layer of MCD12Q2 data (Gray et al., 2019), to 1.26, as evaluated by Wu et al. (2018). The statistics-based CI values estimated by Ray and Foley (2013) are lower than those estimated based on remote sensing data including the GCI30 and those estimated by VIP4 and Wu et al. (2018) using AVHRR satellite observation data. The main reason is that statistics-based CI could not exclude the fallow land area as the agriculture statistics is usually lack of statistical information on fallow land while fallow land could be identified using remote sensing (Zhang et al., 2014a; 2014b) and excluded when generating satellite-based CI products. Our CI is also less than those CI products derived from AVHRR (VIP4 and Wu et al. (2018)). On the one hand, the actual harvest frequency estimated by Wu et al. (2018) might overestimate the annual harvest areas and accordingly overestimate the cropping intensity because they may ignore the presence of fallowed cropland. Each pixel of cropland was assigned to either a single cropping or double cropping category and fallow pixels was not considered which will result in a higher CI (Wu et al., 2018). On the other hand, GCI30 systematically underestimates the cropping intensity when the harvest window is narrow between two growing seasons as a valid phenology season should include both green-up and green-down segments based on the GCI30 algorithm (C. Liu et al., 2020). Interestingly, our CI is exactly the same as the global average from MCD12Q2 for the years 2016 to 2018.

Figure 9 illustrated the differences of statistical annual CI at country scale between GCI30 and four reference datasets. Statistical values of CI at national scale are available in Table S4. National statistical CI values derived from GCI30 are in general close to that of MCD12Q2 and VIP4. The differences over a large proportion of countries were within ±0.3 ranges between GCI30 and those two products, mostly in Asia and Southern Africa. GCI30 and SACRA also presents similar patterns of CI at national scale, especially in Asia. GCI30 presents higher CI values in Central Europe, Southeast Asia Islands, as well as Canada, Brazil and Mexico. In contrast, positive difference values of cropping intensity were commonly observed all over the world as presented by the GCI30 – R&F map. Lower CI values are only observed in few countries in Africa, Asia and Southern America.

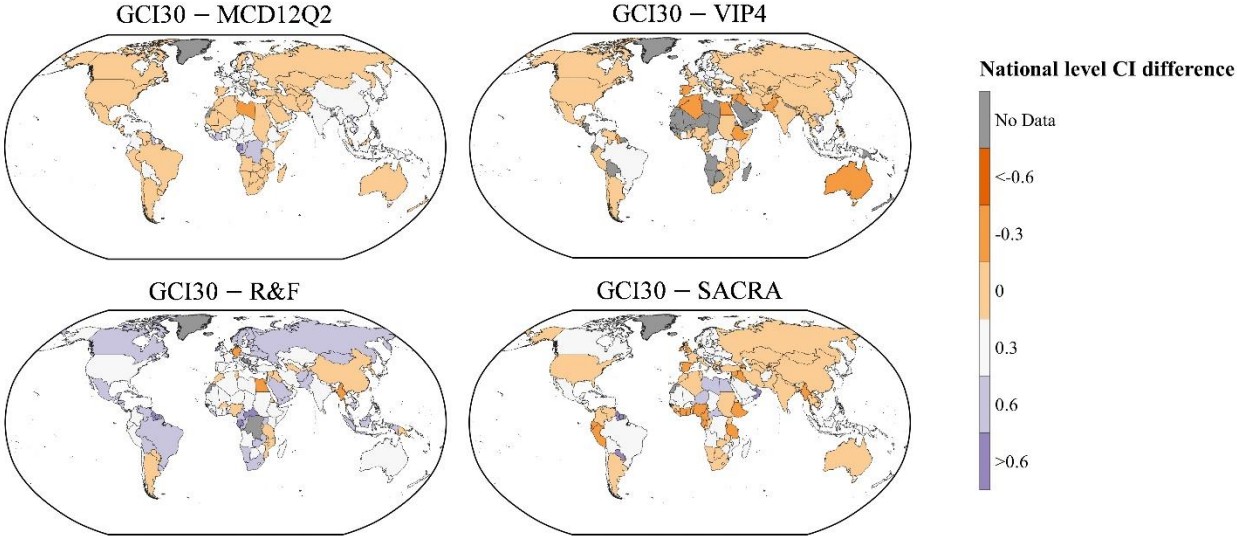

Figure 9: Statistics of annual CI differences at national level between GCI30 and four existing products. GCI30 – MCD12Q2 represents the differences between GCI30 and "NumCycles" layer of MCD12Q2; GCI30 – VIP4 represents the differences between GCI30 and "Number of Seasons" layer of VIP4; GCI30 – R&F represents the differences between GCI30 and harvest frequency by **Ray and Foley (2013)**; GCI30 – SACRA represents the differences between GCI30 and CI by **Kotsuki and Tanaka (2015)**

This study further compared the spatial pattern of the global CI difference between GCI30 and the two global land surface phenology products (MCD12Q2 and VIP4) at pixel level, as displayed in Figure 8. Overall, all three products revealed consistent CI estimations across continents, with zero-difference pixels reaching 79% or higher and the majority of CI differences ranging from -1 to 1. Spatially, positive CI difference values were commonly found in Southeast Asia, the Indian subcontinent and some parts of Europe. Negative CI difference values, on the other hand, were mainly detected in North and South America. There were also discrepancies when these two phenology products were used as the baselines. Referring to MCD12Q2, there were many pixels showing positive values, especially in Africa and mainland China. However, the opposite tendency was observed using VIP4, which exhibited vast negative pixel distributions in Europe and North China.

To further explore how the CI difference varied over space, we selected four 15° × 12° subregions (North America, South America, South Asia and East Asia, which are labelled A, B, C and D, respectively, in Figure 8) that were representative of the global diversity of crop species, climate types and management conditions. In general, substantial variations were detected through these spatially explicit maps. The strongest agreement between GCI30 and MCD12Q2 was found in East Asia (83% of zero values), followed by North America and South Asia, with over 75% agreement. The lowest agreement level occurred in South America, where 32% of the GCI30 estimations showed positive or negative CI differences compared to the MCD12 output. Comparatively, the significant differences and corresponding spatial distributions between the GCI30 and VIP4 outputs had a low level of agreement, although the percentages of pixels with zero difference reached 50% or higher for all subregions. Specifically, three out of the four subregions had at least one-fifth of the pixels featuring negative CI differences. The largest negative disagreement was detected in East Asia, where 41% of the total cropland area had negative values, while North America and South

America also had considerable negative proportions. Finally, in South Asia, the positive and negative pixel
percentages were almost equal, i.e., half and half. Neither MCD12Q2 nor VIP4 should be considered as ground truth.
In fact, the reliability of these two land surface phenology products, especially VIP4, is affected by several factors,
including a coarse spatial resolution, temporal mismatch and algorithm structure differences when compared to GCI30.
Taking East Asia as an example, the "cross-year season cycle" phenomenon (C. Liu et al., 2020) caused by winter
wheat planting could lead to one more "partial growing season" being detected by VIP4 (Didan and Barreto, 2016),
which largely explains why the CI difference between GCI30 and VIP4 shows an outstanding underestimation pattern.

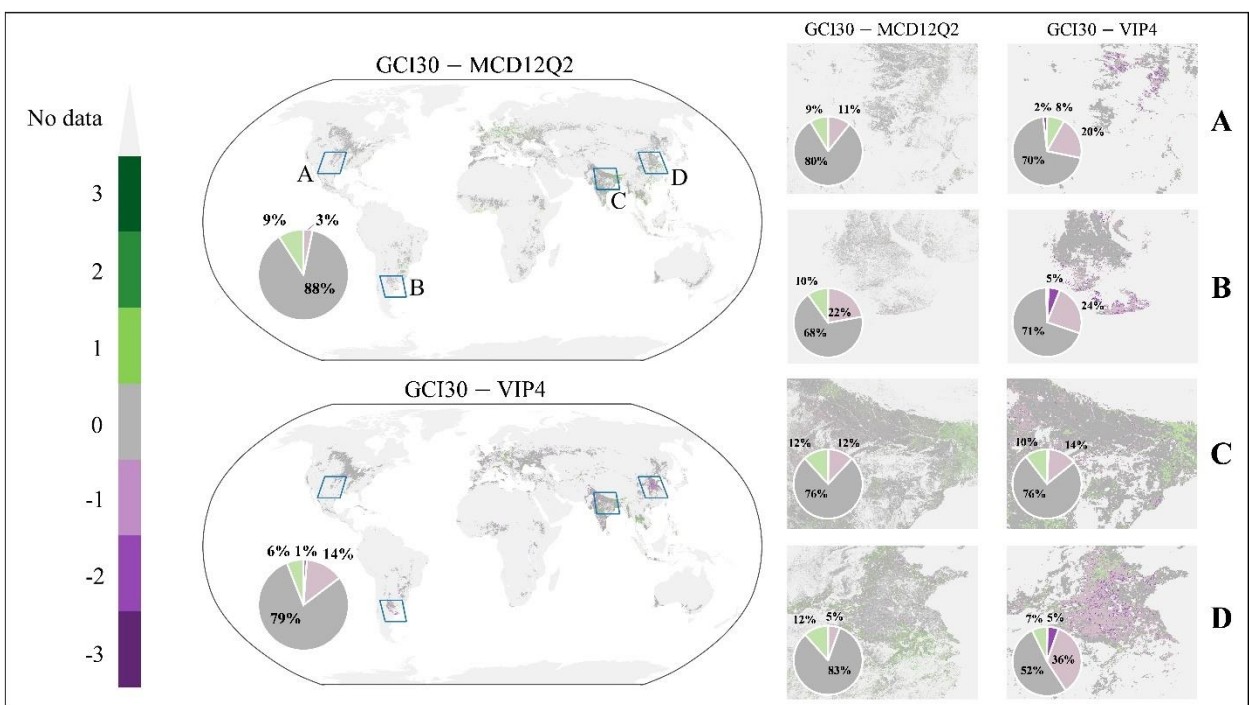

**Figure 10: Spatial patterns and statistics of the CI differences among GCI30, MCD12Q2, and VIP4.**
**3.4 Advantages and limitations of GCI30**
As a global 30-m product, GCI30 depicts the worldwide diversity of agricultural land use intensity in a spatially
explicit manner that has not been fully revealed by existing studies or datasets. Given the CI distribution with a fine
spatial resolution, GCI30 is associated with reduced uncertainties caused by the mixed pixel effect. In addition to the
improvement of mapping accuracy, GCI30 has the potential to monitor landscape-scale cropping practices on
fragmented land parcels by smallholders, which comprise over half of the rural populations in developing nations that
are most vulnerable to food security and environmental challenges (Morton et al., 2006; Jain et al., 2013; Lowder et
al., 2016; C. Liu et al., 2020). Compared with the generalizable crop phenophase pattern, the GCI30 algorithm is not
only efficient in mapping the CI distribution across various AEZs but is also flexible enough to be improved with
updated data inputs. For example, the Harmonized Landsat and Sentinel-2 surface reflectance dataset (Claverie et al.,
2018), with a 5-day revisit interval and a 30-m pixel size, is expected to enhance the global CI mapping performance
once its worldwide coverage is ready. The successful production of GCI30 on the GEE platform illustrates a paradigm

of mapping farming practices that is globally consistent and locally relevant using state-of-the-art cloud computing resources (Lewis et al., 2017; Amani et al., 2020; Tamiminia et al., 2020). It inspires future global fine-scale agricultural research that was previously not applicable.

A large number of natural factors and anthropogenic drivers are related to CI at the planetary scale. Accuracy assessments show that GCI30 explained over 91% and 56% of the sample variations examined by RDsat and RDsite, respectively (Figure 3). The errors of GCI30 could be related to the uncertainties of input data and limitations of the algorithm. The reliability of the cropland extent is a major factor constraining CI mapping performance. To minimize this effect, we integrated an ensemble of 10 land cover or specific cropland layer products for acquiring global cropland extent at a 30-m resolution for 2016–2018. Despite the high overall accuracy of the generated cropland extent, classification errors still exist, especially in some regions of Africa and Asia where small cropland patches are mixed with other land covers (Gong et al., 2013; Xiong et al., 2017). We follow the definition of cropland to select a subset of classes of a layer that best fit in the definition for each of the land-cover/cropland products. Even through, the inconsistency among the 10 land cover or specific cropland layer products still exists. The first issue is greenhouse farming which is included in the cropland class in the FROM-GLC. However, the GCI30 product excluded the greenhouse pixels as CI of greenhouse crops are detected as zero cropping monitored by remote sensing. The second concern is the perennial woody crops such as orchards and vineyards from NLCD. As the NLCD data was only used for Alaska region, it will have very limited impact on the integrated global cropland layer and accordingly minor effect on GCI30. On the other hand, as no single product has yet been shown to be consistently accurate in representing cropland distribution, our approach by integrating different dataset is still better than relying on a single source of land cover or cropland layer (Fritz et al., 2015).

The GCI30 algorithm depends heavily on crop phenological information derived from the time series of vegetation indices. We found that the spatial pattern of the invalid observation count of the 16-day harmonized TOA reflectance composite (Figure S2) matched well with those of RDsat sample bias in some cloudy regions, such as the Gulf of Guinea and East African Highlands (Figure 4), indicating that the performance of GCI30 may be limited in areas suffering from unfavourable weather conditions or extreme seasonal imbalances of clear observations. In particular, the presence of clouds in early and middle of agricultural growing season is preventing optical remote sensing satellites from accurate agricultural applications including cropping cycles detection (Whitcraft et al., 2015; Nabil et al., 2020). Thus, it is reasonable to use the proportion of the invalid number of 16-day composite during 2016-2018 as a quality indicator of GCI30 product. Lower data qualities were observed in Amazon, western Africa, South & Southeast Asia as well as south and southwest China than other regions due to the high cloudy frequency (Figure S2). Although the cloud frequency is relatively low in Western Russia and Central Europe compared with the above cloud-prone regions (Whitcraft et al., 2015), the data quality is also low mainly due to the snow cover in winter and spring. We further evaluated the uncertainty of the GCI30 at global scale. In general, the places with high uncertainty coincided with the cloud-prone regions which might be a resultant of high invalid satellite observations (Figure S2, S3). The fragmented agricultural fields and complex farming practices in the regions including Western Africa, South and Southeast Asia, and East African Highlands (Fritz et al., 2015) further broaden the uncertainty. In Argentina, the cropland field size is

large, and the cloud presence is less frequency. However, large bias of cropping cycles and high uncertainties were commonly observed (Figure 4, 5) which might be attributed to the omission of the poor crops stressed by the severe drought in 2017-2018 agricultural year (Rivera et al., 2021). Rice paddies are fundamentally different from non-flooded croplands, which affects CI mapping performance. We designed a specific rice paddy CI identification approach by considering the influence of the "flooding/transplanting phase". While promising, its application was limited due to the lack of a specific rice paddy layer. Therefore, more improvements can be included, such as integrating SAR data time series for more accurate flood signal detection (Singha et al., 2019).

Additionally, it is noteworthy that the GCI30 product provides insight only into the current actual cropping intensity; however, it is not linked to the potential cropping cycles. To assess the CI gaps between potential and actual situations, climate models could be introduced to simulate the potential cropping cycles under long-term average weather conditions. The proposed method can be readily applied to other years to retrieve long-term CI maps, which will fill in the knowledge gaps of decades of long cropping practices and interannual variations (Iizumi and Ramankutty, 2015). Such information is key to improving our understanding of the CI response to climate in a more granular manner.

**4 Data availability**

The GCI30 product is available on Harvard Dataverse: https://doi.org/10.7910/DVN/86M4PO (Zhang et al., 2020). It is the first 30-m resolution CI dataset covering a global extent. The GCI30 product was tiled into 504 files in GeoTIFF format with geographic projection. To be precise, the spatial resolution of the product is 0.00026949459 degrees. Each GCI30 tile encompasses an area of 10 degree × 10 degree and is named in the following format: 'Cropping_Intensity_30m_2016_2018_$regions$.tif'. The "regions" in the file name are determined as follows: N/S (Northern Hemisphere or Southern Hemisphere) followed by a two-digit latitude label of the tile's top-left corner, and E/W (Eastern Hemisphere or Western Hemisphere) followed by a three-digit longitude label of the tile's top-left corner. Each GeoTIFF file includes two layers. The first layer is the average CI during the three years from 2016 to 2018, with the noData value or masked areas assigned as -1. The valid values for the first layer are 1, 2, and 3, representing single cropping, double cropping or triple cropping, respectively. The second layer is the TNCC from 2016 to 2018 with a noData value or masked areas assigned to -1. The continuous cropping type or the number of cropping cycles larger than 3 per year is assigned as 127 in the abovementioned two layers. We also included a shapefile of the tiles named 'CroppingIntensity_tiles_shapefile.rar' in the repository so that users could easily find their target tiles. The GCI30 product was generated on the GEE platform using JavaScript language developed by the authors. The GEE script as well as the auxiliary data of GCI30 algorithm as an illustration for one tile is open to all potential users and available at https://code.earthengine.google.com/64f569c03f8fd633a896a3ec6f56b89a.

**5 Conclusions**

In this study, we utilized multisource remote sensing data, including Landsat, Sentinel-2 and MODIS data, to produce the first 30-m CI map at a global scale. Based on the phenophase-based mapping framework, GCI30 identified CI by

enumerating the transition points between growing and non-growing periods. To improve the CI mapping performance on flooded rice paddies, we specifically considered the influence of the "flooding/transplanting signal" on the created phenophase profile. Accuracy assessments and inter-comparisons with existing land surface phenology products suggested that GCI30 was reliable across different climate zones and cropping systems. According to GCI30, we estimated that the global average CI was 1.05 during 2016–2018. We found that single-cropping systems occupied more than 80% of the world's cropland extent, while multiple-cropping practices were more commonly observed in South America and Asia than on other continents. National and AEZ-level statistics demonstrated the joint influence of natural and anthropogenic drivers in controlling CI spatial patterns in most areas of the world. We concluded that the new GCI30 dataset provided improved estimates of global CI in a spatially explicit manner that has not been fully captured by previous studies or products and thus can serve to fill data gaps for promoting sustainable agriculture and achieving SDGs.

**Authors contribution**. MZ and CL was responsible for the experimental design, data processing and analysing, paper preparation, revision, and presentation. BW contributed to the conceptual design, reviewing and revising the paper, funding acquisition, and project administration. GH provided support in experimental design, funding acquisition, and project administration. ST, QZ was responsible for data processing and analysing, reviewing and revising the paper. HZ, MN, FT, JB, AB, AE, NY, ZW and YL contributed to the validation data collection, performed data processing and helped in paper revision.

**Competing interests**. The authors declare that they have no conflict of interest. The funders had no role in the decision to publish the results.

**Acknowledgements**. This research was supported by National Key R&D Program of China (2016YFA0600302; 2016YFD0300608; 2019YFE0126900), the Key Collaborative Research Program of the Alliance of International Science Organizations (ANSO-CR-KP-2020-07), the Strategic Priority Research Program of the Chinese Academy of Sciences (XDA19030201), National Natural Science Foundation of China (41861144019 and 41561144013), the Science and technology Project in Guangzhou (202102020584), the Qinghai Science and Technology Plan (2019-SF-155), Microsoft AI for Earth, the American Association of Geographers, and the Group on Earth Observations (GEO) Community Activity Global Ecosystems and Environment Observation Analysis Research Cooperation (GEOARC). The authors sincerely thank Dr. Dongdong Kong for his kindly sharing of the external repositories for running the WHITTAKER smoothing. The authors would like to thank the editor and reviewers for their constructive and insightful comments which help improving the quality of the paper. We also thank all data providers that have been used in this study. Special thanks go to the Google Earth Engine platform and its staff members and user communities.

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
