# Peer review of "GCI30: a global dataset of 30-m cropping intensity using"

_Earth System Science Data, 2021_

## Referee Comment (RC1)

**Review**

**Earth System Science Data**

**Title: GCI30: a global 1 dataset of 30-m cropping intensity using multisource remote sensing image**

**Manuscript ID: Essd-2021-86**

Based on the method tested in China and in a global framework (Liu et al., 2020, Liu et al 2020), this paper developed a global, spatially continuous cropping intensity map at a 30-m resolution (GCI30) using multi-resource satellite data from 2016 to 2018. Accuracy assessments were conducted with visually interpreted validation samples from Geowiki and in situ observations from the PhenoCam network, and they showed reasonably good agreement. The authors further carried out both statistical and spatial comparisons of GCI30 with 6 existing global CI estimates. They also explored the spatial heterogeneity of cropping intensity across countries, continents and Agroecological zones.

Indeed cropping intensity is a critical parameter in agricultural system and sustainable intensification in particular. Undergoing a global study like this is a huge project. The global coverage, very fine-resolution (30 m) and the latest time period (2016-18) indeed fills the data gap for achieving SDGs. However, I do have a few major concerns and I hope the authors could address them to make this paper not only publishable but also even more solid.

(1) My biggest concern (I guess most readers too) is on the method of estimating cropping intensity (Section 2.3). This is obviously the core of this paper. The authors seem to take for granted that they could simply adopt Liu et al (2020) method and apply it to produce the global CI. (BTW, there are two Liu et al (2020), you should specify exactly which one you are referring in the paper). While I acknowledge the good quality of a peer-reviewed paper (or papers if you refer to both papers), there are at least two concerns: one is that what is the major contribution of this paper, or putting it more bluntly how to justify your publishing another paper if you already published two papers: one on China case study, another one on a global CI framework. You need to justify that. The other concern is that global cropping system has much more spatial heterogeneity than your China, or a few regional (in your global paper) cases. For example, subsistence agriculture in Africa (e.g. slash and burn) may include quite a few crops/vegetables within a year, or have fallow period extending multiple years. For many smallholder farmers, your 30m resolution is also too coarse This would not eliminate the mixed pixel problem you cited as one of the big advantage of a fine resolution. In addition, I don't know how Nfc(False crop cycle) is estimated in your method (Page 6, Line 6-9).

(2) The authors divide the croplands into two categories by different mapping method, i.e. non-flooded cropland and flooded rice paddy. Due to the transplanting, flooded rice paddy is treated differently. Again the cropping system is quite diverse, there may have other cropping patterns or farming practices which also need special treatments. For example, the inter-cropping/mixed cropping of a staple crop with a pulse crop (e.g. millet and cowpea) in South Asia, and Sub-Sahara Africa is widely present. I suspect their vegetation indices would also be hard to distinguish and also need special treatment?

(3) One of the important inputs is cropland extent. The authors integrated an ensemble of multiple land cover/cropland layer products. While I applaud the authors' effort of mix and match to try

to get the best available cropland extent globally, such an approach would create another problem of data consistency (e.g. different products even define cropland differently. Orchards are cropland? Plantain or coffee trees?).  I suggest the authors look into Dr. Steffen Fritz work on global cropland. (You used Dr. Fritz' Geowiki datasets and I assumed you are familiar with his work).

(4) Limited reference samples. The authors constructed two independent reference datasets, namely RDsat and RDsite, to evaluate the GCI30 performance. RDsat has 3744 sample records and RDsite has only 40. I understand the difficulty of obtaining the reference samples, particularly in a global study. And yet less than 4000 observations in a hugely diverse cropping system in the world is still quite limited.

In addition, there are many minor issues with expressions, or missing items. I listed some of them here:

(1). Page 4 line 16-17: is there any reference to explain the gap-filling method?

(2). Page 4 line 17: there are four reasons for invalid observation mentioned above (Page 4 line6), but here, authors just list one reason (i.e. clouds) for data gaps. In addition, "vacancy of cloud-free Landsat/Sentinel-2 observation", such expression may cause misunderstanding, whether "cloud-free" means satellite images without clouds or satellite images which were masked by mask algorithm?

(3). Page 5 line 18: add references for GCC

(4). Page 6 line 19: in reference Ding et al. 2020, it is more than 12% instead of 12%.

(5). Fig. 2:

   A. Please modify the font size of the horizontal axis label of Fig.2(a) or Fig.2(b) to ensure that the two graphs have the same font size.
   B. Please modify the vertical axis scale interval of Fig.2(a) or Fig.2(b) to ensure that the two graphs are tidier.
   C. "original phase" in Fig2(a), "flooding signal" in (b), "final phase" in (c), these three dashed polylines are all trapezoidal. But I think rectangular polylines can better represent different phases and transition points.

(6). Fig. 5: please indicate the unit for the Area in the bar charts.

(7) Page 11, Line 14: I don't see this (Wu et al., 2021) in your reference list

(8). Page 12 line 16: shouldn't it be "top 9 countries"?

(9). Page 13 line 7: shouldn't it be "South America"?

(10). Page 14 line 14: change "to" to "two"?

(11). Page 14 line 17: title of table 2. Shouldn't it be "four different studies"?

(12). Page 15 line 15: according to Fig.8(B), shouldn't it be 32% (10%+22%)?

(13). There are many mistakes in reference section and in-text citations (almost all citation formats are incorrect), please check and modify according to the journal reference format requirements.

https://www.earth-system-science-data.net/submission.html#references

---

## Author Comment (AC1)

Thanks for your comments and question. We agree that the cropping season is not completely in line with the calendar year. When detecting cropping intensity for a specific year, the result might be more biased due to the cross-year cycles, which is also addressed in Gray et al. (2014). To avoid such issue, our approach integrated three continuous calendar years to detect cropping cycles rather than one calendar year. This is the main reason in our manuscript to calculate 2016-2018 averaged cropping intensity to reduce the uncertainty resulted from cross-year growing cycles. Using the green chromatic coordinate (GCC) (Richardson et al. 2018) observations at the Shangqiu site (located in North China) as an example (Figure C1), the last growing cycle spans from Nov. 2018 through wintering period to early June 2019. Thus, adopting a multi-year metric would be "safer" to evaluate our outputs, although the CI information is not for a specific year.

[Figure]

Figure C1. Daily GCC time series of Shangqiu site derived from the PhenoCam dataset.

Reference:
Gray, J., Friedl, M., Frolking, S., Ramankutty, N., Nelson, A. and Gumma, M.K., 2014. Mapping Asian cropping intensity with MODIS. IEEE Journal of Selected Topics in Applied Earth Observations and Remote Sensing, 7(8), pp.3373-3379.
Richardson, Andrew D., Koen Hufkens, Tom Milliman, Donald M. Aubrecht, Min Chen, Josh M. Gray, Miriam R. Johnston, et al. 2018. "Tracking Vegetation Phenology across Diverse North American Biomes Using PhenoCam Imagery." Scientific Data 5 (1): 180028. https://doi.org/10.1038/sdata.2018.28.

---

## Author Comment (AC4)

Based on the method tested in China and in a global framework (Liu et al., 2020, Liu et al 2020), this paper developed a global, spatially continuous cropping intensity map at a 30-m resolution (GCI30) using multi-resource satellite data from 2016 to 2018. Accuracy assessments were conducted with visually interpreted validation samples from Geowiki and in situ observations from the PhenoCam network, and they showed reasonably good agreement. The authors further carried out both statistical and spatial comparisons of GCI30 with 6 existing global CI estimates. They also explored the spatial heterogeneity of cropping intensity across countries, continents and Agroecological zones. Indeed cropping intensity is a critical parameter in agricultural system and sustainable intensification in particular. Undergoing a global study like this is a huge project. The global coverage, very fine-resolution (30 m) and the latest time period (2016-18) indeed fills the data gap for achieving SDGs. However, I do have a few major concerns and I hope the authors could address them to make this paper not only publishable but also even more solid.

**Response:** We would like to thank the Referee for the constructive suggestions that help significantly improve the research and the quality of this work. We revised the manuscript according to these comments, and below we provided our detailed responses to the points raised in the supplement.

**Major comments:**

*1 My biggest concern (I guess most readers too) is on the method of estimating cropping intensity (Section 2.3). This is obviously the core of this paper. The authors seem to take for granted that they could simply adopt Liu et al (2020) method and apply it to produce the global CI. (BTW, there are two Liu et al (2020), you should specify exactly which one you are referring in the paper). While I acknowledge the good quality of a peer-reviewed paper (or papers if you refer to both papers), there are at least two concerns: one is that what is the major contribution of this paper, or putting it more bluntly how to justify your publishing another paper if you already published two papers: one on China case study, another one on a global CI framework. You need to justify that. The other concern is that global cropping system has much more spatial heterogeneity than your China, or a few regional (in your global paper) cases. For example, subsistence agriculture in Africa (e.g. slash and burn) may include quite a few crops/vegetables within a year, or have fallow period extending multiple years. For many smallholder farmers, your 30m resolution is also too coarse This would not eliminate the mixed pixel problem you cited as one of the big advantage of a fine resolution. In addition, I don't know how Nfc(False crop cycle) is estimated in your method (Page 6, Line 6-9).

**Response:** Thank you for raising the point which encourages us to clarify our main methodology and emphasize the contribution of this manuscript. The paper we refer our methodological framework to is the one titled "A new framework to map fine resolution cropping intensity across the globe: Algorithm, validation, and implication" published in Remote Sensing of Environment by C. Liu et al. (2020). In this cited paper, eight 10 by 10-degree regions across the terrestrial world were selected to test the performance of the main algorithm for mapping cropping intensity. Estimated cropping intensity was validated also for the eight regions. Thus, C. Liu et al. (2020) paper as a pilot study focused on justifying the robustness and solidness of the algorithms within the designed framework.

Moving a step forward, in this manuscript, the contributions are to substantially improve the algorithm of mapping cropping intensity, practically applying the framework for the entire world (i.e., all cropland over the terrestrial surface), and publicizing the global product to serve the research and education community. We generated the layer of cropping intensity at 30m resolution for all cropland during 2016-2018, and packaged the global product and publicized it to the public (non-commercial and main purposes for research, education, and policy evaluation). To our best knowledge, this is so far the most updated and latest product on cropland-specific intensity at 30m for the globe, which can be beneficial for a variety of research and practical uses key to sustainable development.

For the cropping system diversity issue, we agree that accounting for the spatial heterogeneity is challenging, and what we are doing is to develop a simple but effective approach for global CI mapping at fine resolution. In fact, our pilot study did include an Africa region as a part for method validation (C. Liu et al. 2020). As expected, this region exhibited reasonable yet relatively low accuracy compared to other regions. As the first version GCI30, we accepted the tradeoff between accuracy and efficiency, and we will continue improve our technical framework to update later GCI30 versions and expand the temporal coverages in the future. Spectral mixture is a common uncertainty source when applying satellite image mapping, and its influence is related to the raster pixel size. A major advantage of GCI30 lies in its 30-m spatial resolution, which is much higher than previous dataset. Though the spectral mixture issue still exists at 30m, we would argue that the mixture effects is lower than the existing research on cropping intensity mapping and the previously available products. We added some description of the mixed pixels in the result and discussion part according in **Section 3.1**.

For the Nfc exclusion, we added sentences in the revised manuscript explaining our procedure in **Section 2.3.1**.

*2 The authors divide the croplands into two categories by different mapping method, i.e. non-flooded cropland and flooded rice paddy. Due to the transplanting, flooded rice paddy is treated differently. Again the cropping system is quite diverse, there may have other cropping patterns or farming practices which also need special treatments. For example, the inter-cropping/mixed cropping of a staple crop with a pulse crop (e.g. millet and cowpea) in South Asia, and Sub-Sahara Africa is widely present. I suspect their vegetation indices would also be hard to distinguish and also need special treatment?

**Response:** Thanks for raising the issue on clarification. We agree with the comment that global cropping systems are highly diverse due to factors including climate, policy, and socioeconomic conditions. 1) Instead of pursuing algorithmic consideration for each cropping type, this study aims to use a general scheme for creating GCI30 that is efficient and representative of the major cropping types worldwide. 2) We treated flooded paddy rice differently because of two reasons. The first reason is that rice is a major crop, especially for many developing countries. The second reason is that there are successful applications of paddy rice transplanting characteristics (Xiao et al. 2005; Dong et al. 2015; 2016), which can guide our CI mapping in this study. 3) Identification of inter/mixed cropping using satellite remote sensing is extremely challenging. Since the proposed approach is pixel-based, it reflects the composition of all cropping system within that pixel. Unfortunately, we did not have *in situ* samples of inter/mixed cropping in South Asia or Sub-Sahara Africa as mentioned by the Referee. Here we provide a typical example among others in Northeast China to test this intercropping issue, as shown in the following figure. Fig. R1 shows a cropland pixel located in northeast of China where maize and soybean are intercropped. Although maize and soybean are simultaneously planted in alternating rows of the same pixel, satellite NDVI time series is still able to reflect the greening/browning cycles of the entire 30×30 m extent. In spite of these promising results, it should be recognized that for some extreme cases, in which two or more crops have totally different phenological features, our method may be less reliable. We added this information in the revised manuscript.

[Figure]

Figure R1 a demo showing NDVI time series of a inter/mixed cropping system

*Reference 1: Xiao, Xiangming, Stephen Boles, Jiyuan Liu, Dafang Zhuang, Steve Frolking, Changsheng Li, William Salas, and Berrien Moore III. 2005. "Mapping Paddy Rice Agriculture in Southern China Using Multi-Temporal MODIS Images." Remote Sensing of Environment 95 (4): 480–92.*

*Reference 2: Dong, Jinwei, and Xiangming Xiao. 2016. "Evolution of Regional to Global Paddy Rice Mapping Methods: A Review." ISPRS Journal of Photogrammetry and Remote Sensing 119: 214–27.*

*Reference 3: Dong, Jinwei, Xiangming Xiao, Michael A Menarguez, Geli Zhang, Yuanwei Qin, David Thau, Chandrashekhar Biradar, and Berrien Moore III. 2016. "Mapping Paddy Rice Planting Area in Northeastern Asia with Landsat 8 Images, Phenology-Based Algorithm and Google Earth Engine." Remote Sensing of Environment 185: 142–54.*

*3 One of the important inputs is cropland extent. The authors integrated an ensemble of multiple land cover/cropland layer products. While I applaud the authors' effort of mix and match to try to get the best available cropland extent globally, such an approach would create another problem of data consistency (e.g. different products even define cropland differently. Orchards are cropland? Plantain or coffee trees?). I suggest the authors look into Dr. Steffen Fritz work on global cropland. (You used Dr. Fritz' Geowiki datasets and I assumed you are familiar with his work).

**Response:** Thanks for this valuable comment on the integration of different existing products. The cropland definition adopted in our research is based on the concept presented by the Joint Experiment of Crop Assessment and Monitoring (JECAM)

network which was created by the Group on Earth Observation Global Agriculture Monitoring Community of Practice. The JECAM network has adopted a shared definition of the cropland that matches the Food and Agriculture Organization's (FAO) Land Cover Meta Language. The general definition of annual cropland (including area affected by crop failure) is a piece of arable land that is sowed or planted at least once within a 12-month period. The annual cropland produces an herbaceous cover and is sometimes combined with some tree or woody vegetation. One exception is the sugarcane plantation and cassava crop, which are included in the cropland class although they have a longer vegetation cycle and are not yearly planted.

In our research, we integrated 10 existing global, regional, and national land-cover maps, or cropland dataset (listed in the table S1 in the supplementary document) to delimit the global cropland extent while masking out irrelevant non-cropland pixels for the period of 2016–2018 (Figure 1). Although variations of classification systems among different products exist, a subset classes of those land cover/cropland layer products were selected to best fit into the cropland definition. We revised the manuscript in **Section 2.1.1** accordingly.

In our case, there are two known exceptions in the integration. The first is greenhouse farming which is included in the cropland class in the FROM-GLC by the definition. However, the GCI30 product excluded the greenhouse pixels as CI of greenhouse crops are detected as zero cropping monitored by remote sensing. The second is the perennial woody crops such as orchards and vineyards from NLCD. As the NLCD data was only used for Alaska region, it will have very limited impact on the integrated global cropland layer and accordingly minor effect on GCI30. On the other hand, as no single product has yet been shown to be consistently accurate in representing cropland distribution, our approach by integrating different dataset is still better than relying on a single source of land cover or cropland layer which already pointed out by Fritz et al. (2015). We added those descriptions in section 3.4 to further discussion the uncertainty resulted from the integration of different dataset.

As suggested, we also revised the table S1 in the supplementary document to include subclasses selected for the integration from each dataset and the corresponding definitions.

*Reference*:

*Fritz, S., See, L., McCallum, I., You, L., Bun, A., Moltchanova, E., Duerauer, M., Albrecht, F., Schill, C., Perger, C. and Havlik, P., 2015. Mapping global cropland and field size. Global change biology, 21(5), 1980-1992.*

*4 Limited reference samples. The authors constructed two independent reference datasets, namely RDsat and RDsite, to evaluate the GCI30 performance. RDsat has 3744 sample records and RDsite has only 40. I understand the difficulty of obtaining the reference samples, particularly in a global study. And yet less than 4000 observations in a hugely diverse cropping system in the world is still quite limited.

**Response:** Thank you for this comment. As you mentioned, it is a challenging task to construct a reliable reference dataset for GCI30 evaluation. This is because 1) there is still no product can be directly used for global CI mapping assessment; 2) the identification of CI value by visual interpretation requires not only the location information, but also precisely judging the number of growing seasons, which is time consuming and laborious; 3) the reference dataset should represent the diversity of global cropping systems. For the first issue, it is the primary reason for us to build new reference sets. For the second issue, we have seven remote sensing experts (listed as coauthors) checked all collected points, and only well-interpreted points with high-level confidence were kept. For the third issue, we adopted a stratified sampling approach to ensure that RDsat was geographically representative across the globe. Moreover, we utilized all available PhenoCam cropland in-situ data although its spatial representative is somehow limited. In summary, multiple efforts were made to make our evaluation as solid as possible, and we agree with the comment that the current RDsat and RDsite data are far away from perfect, which needs further cooperation and study in the future.

**Minor comments:**

*1 Page 4 line 16-17: is there any reference to explain the gap-filling method?

**Response:** We added sentences in **the last paragraph in Section 2.1** explaining the adopted gap-filling method: "In particular, the coarse MODIS datasets were resized to 30-m using the bicubic interpolation method. Then an empirical linear function was built for each pixel to bridge the data records of MODIS and Landsat/Sentinel-2, and missing data gaps were filled with the resampled, transformed MODIS data (labelled as MODIS modelled) (Liu et al. 2020). If there is no valid data from either Landsat/Sentinel-2 or MODIS, temporally adjacent (within 48-day) cloud free LANDSAT/Sentinel-2 observations were used to determine the filling value (labelled as interpolated)."

*Reference: Liu, Chong, Qi Zhang, Shiqi Tao, Jiaguo Qi, Mingjun Ding, Qihui Guan, Bingfang Wu, et al. 2020. "A New Framework to Map Fine Resolution Cropping Intensity across the Globe: Algorithm, Validation, and Implication." Remote Sensing of Environment 251: 112095. https://doi.org/10.1016/j.rse.2020.112095.*

*2 Page 4 line 17: there are four reasons for invalid observation mentioned above (Page 4 line6), but here, authors just list one reason (i.e. clouds) for data gaps. In addition, "vacancy of cloud-free Landsat/Sentinel-2 observation", such expression

may cause misunderstanding, whether "cloud-free" means satellite images without clouds or satellite images which were masked by mask algorithm?

**Response:** Thank you for this comment! We modified the sentence to "We also used the MOD13Q1 NDVI/EVI product and MOD09A1-derived LSWI in our study to fill data gaps caused by the vacancy of Landsat/Sentinel-2 observations that were removed by the Fmask algorithm".

*3 Page 5 line 18: add references for GCC

**Response:** Reference (Richardson et al., 2018) added.

*Reference: Richardson, Andrew D., Koen Hufkens, Tom Milliman, Donald M. Aubrecht, Min Chen, Josh M. Gray, Miriam R. Johnston, et al. 2018. "Tracking Vegetation Phenology across Diverse North American Biomes Using PhenoCam Imagery." Scientific Data 5 (1): 180028. https://doi.org/10.1038/sdata.2018.28.*

*4 Page 6 line 19: in reference Ding et al. 2020, it is more than 12% instead of 12%.

**Response:** Corrected in the revised manuscript.

*5 Fig. 2:

    A. Please modify the font size of the horizontal axis label of Fig.2(a) or Fig.2(b) to ensure that the two graphs have the same font size.

    B. Please modify the vertical axis scale interval of Fig.2(a) or Fig.2(b) to ensure that the two graphs are tidier.

    C. "original phase" in Fig2(a), "flooding signal" in (b), "final phase" in (c), these three dashed polylines are all trapezoidal. But I think rectangular polylines can better represent different phases and transition points.

**Response:** Agreed and we modified Figure 2 in the revised manuscript.

*6 Fig. 5: please indicate the unit for the Area in the bar charts.

**Response:** Modified in the revised manuscript.

*7 Page 11, Line 14: I don't see this (Wu et al., 2021) in your reference list

**Response:** Thanks for this comment. As Wu et al., 2021 is an unpublished citation, we now replace this citation by (Zohaib and Choi, 2020) and rewrite the sentence as follows: "These regions are commonly characterized by warm and humid climates, except for the Nile River basin, in which irrigation has been commonly used to support intensive farming practices (Zohaib and Choi, 2020)."

*8 Page 12 line 16: shouldn't it be "top 9 countries"?

**Response:** We appreciate this comment. Sri Lanka was lost in the list and we fixed this mistake in the revised manuscript.

*9 Page 13 line 7: shouldn't it be "South America"?

**Response:** Thanks and we corrected this bug in the revised manuscript.

*10 Page 14 line 14: change "to" to "two"?

**Response:** Corrected.

*11 Page 14 line 17: title of table 2. Shouldn't it be "four different studies"?

**Response:** Corrected. This table is now moving to methodology part in **section 2.5 Comparison with other global products** to demonstrate which studies and existing products are used for inter-comparison.

*12 Page 15 line 15: according to Fig.8(B), shouldn't it be 32% (10%+22%)?

**Response:** Agreed and corrected.

*13 There are many mistakes in reference section and in-text citations (almost all citation formats are incorrect), please check and modify according to the journal reference format requirements.

https://www.earth-system-science-data.net/submission.html#references

**Response:** Agreed and modified. We have gone through the manuscript and made sure all citations (including the Reference section and in-text citations) correct according to ESSD's requirement.

---

## Author Comment (AC5)

This study combined Landsat, Sentinel-2, and MODIS images to generate a global cropping intensity map at a spatial resolution of 30 meters between 2016 and 2018. This study also validated the cropping intensity map using thousands of pixels from time series satellite images and currently available PhenoCam data. Then this study compared the resultant cropping intensity map with other cropping intensity datasets. This is nice work. The writing and the logic of this manuscript are good. However, I have some comments which may help the authors improve this study.

**Response:** We would like to thank the Referee for the constructive suggestions that help significantly improve the research and the quality of this work. We revised the manuscript according to these comments and provided our detailed responses point to point to address all the issues raised.

*1 Page 1 Line 34-35: This study did not do the relevant analysis to support this conclusion.

**Response:** Agreed and modified. We changed the sentence in the **Abstract** to: "As the first global coverage, fine-resolution CI product, GCI30 is expected to fill the data gap for achieving sustainable development goals (SDGs) by depicting worldwide diversity of agricultural land use intensity."

*2 Page 2 Line 14: Here is the definition of cropping intensity. What is the continuous cropping type you mentioned in the manuscript?

**Response:** Thanks for raising the issue on clarification. We provided the definition of "continuous cropping" in **Section 2.3.1** because it is analogous to other CI categories including single cropping, double cropping, and triple cropping. In our manuscript, continuous cropping is defined as cropping systems having short growing period (CI > 3 for this study) or exhibiting a lower degree of seasonality (e.g., tea plantation).

*3 Page 3 Line 6: You may need to highlight why you combined Landsat, Sentinel-2 and MODIS data somewhere in the Introduction section.

**Response:** Agreed. We added a sentence in **the last paragraph of the Introduction Section** to explain the reason of the combination of multiple data: "We integrated the full archive of Landsat, Sentinel-2 and MODIS data from 2016 to 2018 for constructing seamless spectral time series in order to capture the full cropping cycles,

which is the key for CI identification by segmenting growing and non-growing periods."

*4 Page 3 Line 15-25: According to your Google Earth Engine code, this study included forest, water, and urban mask to help integrate the cropland layer. But I did not find any description in this section.

**Response:** Thanks for pointing out this issue. As you mentioned, the initial version of our GEE code utilized some global urban, forest and water products help mask non-cropland extents (See our RSE paper for more details, C. Liu et al., 2020). However, for this study of GCI30 producing, we improved the procedure of non-cropland masking by using an ensemble of multiple land cover/cropland layer products. In particular, the demo code is located in U.S., where the CDL dataset was used. We have updated our GEE code accordingly which is open to all and available at https://code.earthengine.google.com/3572b843c607c25ba9be876e6f73948e

*Reference: Liu, Chong, Qi Zhang, Shiqi Tao, Jiaguo Qi, Mingjun Ding, Qihui Guan, Bingfang Wu, et al. 2020. "A New Framework to Map Fine Resolution Cropping Intensity across the Globe: Algorithm, Validation, and Implication." Remote Sensing of Environment 251: 112095. https://doi.org/10.1016/j.rse.2020.112095.*

*5 Page 4 Line 4-5: Why not using surface reflectance data?

**Response:** We used top-of-atmosphere (TOA) reflectance because of two reasons. First, within the GEE storage, the Sentinel-2 TOA data have longer temporal coverage (since 2015) than the surface reflectance (since 2017). Moreover, the inter-calibration approach harmonizing Landsat and Sentinel-2 (Chastain et al. 2019) was based on TOA data. As a result, TOA rather than surface reflectance data were used.

*Reference: Chastain, Robert, Ian Housman, Joshua Goldstein, Mark Finco, and Karis Tenneson. 2019. "Empirical Cross Sensor Comparison of Sentinel-2A and 2B MSI, Landsat-8 OLI, and Landsat-7 ETM+ Top of Atmosphere Spectral Characteristics over the Conterminous United States." Remote Sensing of Environment 221: 274–85. https://doi.org/10.1016/j.rse.2018.11.012.*

*6 Page 4 Line 16-17: Landsat and Sentinel-2 are TOA data, and MODIS data are surface reflectance (SR) data if my understanding is right. The reflectance values

from TOA data and SR data should be very different, especially when including the blue band to calculate EVI. Which version of MODIS data are used?

**Response:** 1) Indeed, we only used the MODIS surface reflectance (SR) data (MOD09A1) to generate LSWI layer. The MODIS NDVI/EVI is not used to generate vegetation indices. The MOD13Q1 NDVI/EVI product was used but cannot be directly combined with Landsat and Sentinel-2. We added sentences in the revised manuscript explain how the MODIS data were harmonized and integrated with fine resolution imagery as follows: In particular, the coarse MODIS datasets were resized to 30-m using the bicubic interpolation method. Then an empirical linear function was built for each pixel to bridge the data records of MODIS and Landsat/Sentinel-2, and missing data gaps were filled with the resampled, transformed MODIS data (labelled as MODIS modelled). If there is no valid data from either Landsat/Sentinel-2 or MODIS, temporally adjacent (within 48-day) cloud free LANDSAT/Sentinel-2 observations were used to determine the filling value (labelled as interpolated). 2) For all MODIS data, we used the Collection 6 version MOD13Q1 NDVI/EVI and MOD09A1 to derive LSWI data.

*7 Page 5 Line 1-2: More details are needed for the data gap filling and smoothing. The integrated data is 30 meters, but what is the temporal resolution?

**Response:** Agreed and modified. We added description of data gap filling and smoothing in the revised manuscript. The temporal resolution (16-day) was also given. The text is revised to: "After gap-filling, a weighted Whittaker smoother (Kong et al. 2019) was further adopted to smooth the gap filled time series data. We assigned different weights (1, 0.5, 0.2) to Landsat/Sentinel-2 original observations, MODIS modelled values and interpolation values, respectively. Finally, a dataset of smoothed, seamless image time series of vegetation indices was generated at a spatial resolution of 30-m with a temporal interval of 16-day."

Please see the last paragraph in **Section 2.1.**

*8 Page 5 Line 14: what is the size for these samples?

**Response:** Thanks for this comment. The spatial size of RDsat is 30 m $\times$ 30 m.

*9 Page 5: Line 25: How did you separate flooded and non-flooded croplands?

**Response:** Thanks for the comment. For this study, the flooding/non-flooding information is derived from our cropland extent map. More specifically, ChinaCover and SERVIR land cover products have paddy rice included in their classification systems. Please refer to the Table S1 in the supplementary document. This also explained why the flooding specific method was only applied in China and Lower Mekong River basin.

*10 Page 6 Line 1: Why did you choose the 50% of the NDVI amplitude?

**Response:** This is a good point! We use 50% amplitude because it is in the middle of peak and dent, which maximize the relative distance from the selected point to each. More importantly, it was also adopted by a previous study (Bolton et al. 2020) for separating growing/non-growing periods. We added the reference in the revised manuscript.

*Reference: Bolton, D.K., Gray, J.M., Melaas, E.K., Moon, M., Eklundh, L. and Friedl, M.A., 2020. Continental-scale land surface phenology from harmonized Landsat 8 and Sentinel-2 imagery. Remote Sensing of Environment, 240, p.111685.*

*11 Page 7 Figure 2: This figure is not clear to me. Could you please the ticks for the x-axis?

**Response:** We modified Figure 2 in the revised manuscript.

*12 Page 8 Line 9-25: A brief introduction of these products is needed. Or you can move Table 2 here and add the two phenology datasets in the table.

**Response:** Agreed and modified. We moved Table 2 (as Table 1 in the revised manuscript), offering information of each product.

*13 Page 9 Line 14-25: Could you explain why the underestimation and overestimation happened based on those samples (RDsat)?

**Response:** Thanks for this comment. We added further discussion of potential drivers shaping such a bias distribution using RDsat as follows:

The negative errors could possibly be due to the complexity of some special cropping systems that cannot be fully accounted by our CI mapping method. For example, inter/mixed cropping may lead to shallow troughs in NDVI time series, which makes the 50% NDVI amplitude rule less reliable (C. Liu et al. 2020).

These positive biases could be attributed to the fallow strategy adopted in some Europe countries (Estel et al. 2016). During a fallow cycle, there may exist weeds which are falsely identified as one solid cropping cycle.

Please see the revised text in **Section 3.1**.

*Reference:*

*Estel, S., Kuemmerle, T., Levers, C., Baumann, M. and Hostert, P., 2016. Mapping cropland-use intensity across Europe using MODIS NDVI time series. Environmental Research Letters, 11(2), p.024015.*

*14 Page 10 Figure 4: I could not see the cropping cycles dots.

**Response:** Thanks for this comment. For Figure 4, we use the size of dot indicating the actual total number of cropping cycle(s), and the color representing the prediction biases. Therefore, each point in the map is a combination of two features: different sizes and different colors. We reclarified this in the figure caption.

*15 Page 11 Figure 5: The colors for different cropping intensity types are too close. Please modify the colors.

**Response:** Agreed and modified.

*16 Page 14 Line 2: Why excluding the continuous cropping pixels?

**Response:** The continuous cropping pixels were excluded for mean/std calculation because it will bring in additional uncertainties. According to its definition as added in the last paragraph in Section 2.3.1, there is no explicit CI value for some continuous cropping systems like evergreen cash crops. Therefore, this special cropping type should be removed before statistical analysis.

*17 Page 14 Line 19-20: Why "Wu et al (2018) might overestimate the annual harvest area and …"?

**Response:** Thanks for your question. We realized that we only described the higher CI from Wu et al (2018) but did not explain the reason. We have now revised the manuscript and added an explanation as follows: Wu et al. (2018) might overestimate the annual harvest areas and accordingly overestimate the cropping intensity because they ignored the presence of fallowed cropland. Each pixel of cropland was assigned to either a single cropping or double cropping category which will result in a higher CI.

*18 Page 14 Table 2: Please cite the relevant references. FAO data in 2010 is used for comparison, and the FAO data in 2016-2018 should be better for comparison.

**Response:** Thanks for your comments. We do agree that the FAO data in 2016-2018 should be better for comparison with GCI30 to avoid the uncertainty resulted from temporal differences. However, the FAO data is coming from published paper (Siebert et al. (2010) which used the statistical data up to 2010. Therefore, we did not use FAO data in 2016-2018 for comparison. As cropping intensity derived from FAO data is only the statistical information at global or a certain administrative unit, in our revised manuscript, we dropped the CI using FAO data and focused on the comparison with other products from both statistically and spatially aspects at global scale.

*19 Page 15 Line 7: MDC12Q2 should be MCD12Q2.

**Response:** Agreed and corrected.

*20 Page 16 Line 4: This study "provides insight only into the current actual cropping intensity (Page 17 Line 18)" instead of "agricultural land use management".

**Response:** Agreed and modified. We changed the sentence to: "As a global 30-m product, GCI30 depicts the worldwide diversity of agricultural land use intensity in a spatially explicit manner that has not been fully revealed by existing studies or datasets." It was added to the beginning of **Section 3.4**.

*21 Page 16 Line 7: I agree that the GCI30 generated by this study reduces uncertainties caused by the mixed pixel effect. However, suppose you would like to say you generated a more accurate global CI estimation. In that case, you need to do

accuracy assessment for each cropping intensity product based on their specific definition and the same set of ground reference samples.

**Response:** Agreed. We rewrote this sentence as: "Given the CI distribution with a fine spatial resolution, GCI30 is associated with reduced uncertainties caused by the mixed pixel effect".

*22 Table S1: You may add the spatial resolution for each product.

**Response:** Agreed and added. Please see the revised Table S1.

*23 This manuscript is too long and could be shortened.

**Response:** Thanks for your suggestions. We agree that the manuscript is relatively long. However, the length of our manuscript is reasonable if we compare with some published papers on ESSD. We keep the manuscript long because of two reasons.

1) Our manuscript is the first manuscript describing the global pattern of cropping intensity at 30m resolution. We documented the full methodologies and spatial pattern and statistical characteristics of the derived GCI30 product.

2) We used a couple of pages to compare GCI30 with the existing research and products from both statistical and spatial aspect to make it clear the advantages of our product.

Nevertheless, the authors went through the revised manuscript and dropped the redundant sentences or reorganize the wordings to improve the readability.

---

## Author Comment (AC6)

Dear Authors,

I still have two comments for #5 and #6. Regarding #5, it is not uncertainty. It sounds like quality indicator. Regarding #6, my suggestion is to add a table of CI statistics from different products. It will be of great help to provide such table at the country level in SI. I will send it out for review now to speed the process. But please address these two comments together with comments from reviewers when you have them and if you receive a decision of revision.

**5. The uncertainty or quality of the data should be added in the product. As seen from this paper, the spatial variation of uncertainty is significant.**

**6. A result table of comparison with other studies can be added. It is difficult to evaluate the difference of this product with other studies based on current presentation.**

**Response:** We would like to thank for the constructive comments and suggestions from Topical Editor that help significantly improve the research and the quality of this work. Thanks for your kindness to send our manuscript out for review and open discussion which speed up the process duration even with two comments from you not properly addressed. We further revised the manuscript addressing the two comments, and below we provided our detailed responses to your concerns.

*1 Regarding #5, it is not uncertainty. It sounds like quality indicator.

**Response:** Thanks for your comments. We agree that the proportion of the invalid number of 16-day composite during 2016-2018 is a quality indicator rather an index of uncertainty. The more invalid 16-day composite TOA reflectance observations, the lower quality of the input data and the cropping intensity products are. We modified the quality map by ranking the number of invalid observations from 1 to 10. Regions with zero invalid observations are marked as highest quality at 10 while regions with more than 56 invalid observations are marked as worst quality at 1. Such a quality map is added in the supplementary document (Figure S3).

[Figure]

Figure S3. Data quality map of GCI30 measured as the invalid number of 16-day composite during 2016-2018. Zero indicates lowest data quality while 10 indicates highest data quality.

Moreover, as suggested, a systematic uncertainty analysis was applied referring to a published paper on Earth System Science Data (H. Liu et al., 2020) by interpolation the uncertainty from validation samples' locations to a spatial distribution map. Among the 3744 validation samples, uncertainties are marked as 1 when the absolute values of prediction bias of the cropping intensity are equal or larger than 2, while test sample locations with zero bias are marked as 0. The spatial distribution map of the uncertainty of GCI30 result is generated based on a Kriging interpolation method (Oliver and Webster, 1990) using ArcMap software. The search radius parameter of Kriging interpolation is set to 12 nearby sample units, the other parameters as default. The value of the uncertainty ranges from 0 to 1. A value close to 0 indicates a lower uncertainty, while a value near to 1 indicates a higher uncertainty and a higher possibility of overestimation or underestimation on cropping cycles. We also added the description of the uncertainty method in the methodology part in section 2.4. In the result and discussion part, a global uncertainty map of GCI30 product (Figure 5 in the revised manuscript) is added and the uncertainty of the GCI30 is further discussed in section 3.1 Reliability of GCI30.

[Figure]

Figure 5: Global uncertainty map of GCI30 during 2016-2018, where regions in red represent higher uncertainty and those in blue represent lower uncertainty.

*References:*

*Liu, H., Gong, P., Wang, J., Clinton, N., Bai, Y., and Liang, S.: Annual dynamics of global land cover and its long-term changes from 1982 to 2015, Earth Syst. Sci. Data, 12, 1217–1243, https://doi.org/10.5194/essd-12-1217-2020, 2020.*

*Oliver, M. A., & Webster, R. (1990). Kriging: a method of interpolation for geographical information systems. International Journal of Geographical Information System, 4(3), 313-332.*

*2 Regarding #6, my suggestion is to add a table of CI statistics from different products. It will be of great help to provide such table at the country level in SI.

**Response:** Thanks for this valuable comment. A new figure (Figure 9 in the revised manuscript) is added to show the differences of annual CI statistics between GCI30 and four existing products at country level. We further described the spatial pattern of the variations between our products and existing ones at national scale in section 3.3. Cross comparison with other studies as follows: National statistical CI values derived from GCI30 are in general close to that of MCD12Q2, and VIP4. The differences over a large proportion of countries were within ±0.3 ranges between GCI30 and those two products, mostly in Asia and Southern Africa. GCI30 and SACRA also presents similar patterns of CI at national scale, especially in Asia. GCI30 presents higher CI values in Central Europe, Southeast Asia Islands, as well as Canada, Brazil and Mexico. In contrast, positive difference values of cropping intensity were commonly observed all over the world as presented by the GCI30 – R&F map. Lower CI values

are only observed in few countries in Africa, Asia and Southern America. In the supplementary document, as suggested, we also added a table (Table S4) at country level to compare the CI statistics from our GCI30 and other four different global CI product (the "NumCycles" layer of MCD12Q2, the "Number of Seasons" layer of VIP4, harvest frequency by Ray and Foley (2013) and SACRA product (Kotsuki and Tanaka, 2015)). Please note that the CI statistics is only available for GCI30 and four other global products because only those global products are available or accessible.